# GWAS for quantitative resistance phenotypes in *Mycobacterium tuberculosis* reveals resistance genes and regulatory regions

Maha R. Farhat[1,2], Luca Freschi[1], Roger Calderon[3], Thomas Ioerger[4], Matthew Snyder[5], Conor J. Meehan[6], Bouke de Jong[6], Leen Rigouts [6], Alex Sloutsky[7], Devinder Kaur[8], Shamil Sunyaev [1,9], Dick van Soolingen[10], Jay Shendure[5,11,12], Jim Sacchettini[4] & Megan Murray[13]

Drug resistance diagnostics that rely on the detection of resistance-related mutations could expedite patient care and TB eradication. We perform minimum inhibitory concentration testing for 12 anti-TB drugs together with Illumina whole-genome sequencing on 1452 clinical *Mycobacterium tuberculosis* (MTB) isolates. We evaluate genome-wide associations between mutations in MTB genes or non-coding regions and resistance, followed by validation in an independent data set of 792 patient isolates. We confirm associations at 13 non-canonical loci, with two involving non-coding regions. Promoter mutations are measured to have smaller average effects on resistance than gene body mutations. We estimate the heritability of the resistance phenotype to 11 anti-TB drugs and identify a lower than expected contribution from known resistance genes. This study highlights the complexity of the genomic mechanisms associated with the MTB resistance phenotype, including the relatively large number of potentially causal loci, and emphasizes the contribution of the non-coding portion of the genome.

[1] Department of Biomedical Informatics, Harvard Medical School, Boston, MA, USA. [2] Division of Pulmonary and Critical Care, Massachusetts General Hospital, Boston, MA, USA. [3] Socios en Salud, Lima, Peru. [4] Texas A and M University, College Station, TX, USA. [5] Department of Genome Sciences, University of Washington, Seattle, WA, USA. [6] Department of Biomedical Sciences, Institute of Tropical Medicine, Antwerp, Belgium. [7] Massachusetts Supranational TB Reference Laboratory, University of Massachusetts Medical School, Boston, MA, USA. [8] New England Newborn Screening Program, University of Massachusetts Medical School, Worcester, MA, USA. [9] Department of Genetics, Brigham and Women's Hospital, Boston, MA, USA. [10] National Institute for Public Health and the Environment (RIVM), Bilthoven, The Netherlands. [11] Howard Hughes Medical Institute, Seattle, WA, USA. [12] Brotman Baty Institute for Precision Medicine, Seattle, WA, USA. [13] Department of Global Health and Social Medicine, Harvard Medical School, Boston, MA, USA. Correspondence and requests for materials should be addressed to M.R.F. (email: Maha_Farhat@hms.harvard.edu)

Tuberculosis (TB) remains a major global public health threat. In 2016, there were an estimated 10.4 million TB cases globally and 1.7 million deaths owing to the disease. One of the most-challenging forms of disease is caused by multidrug resistant (MDR) *Mycobacterium tuberculosis*, with a global annual incidence of over half a million cases[1]. The World Health Organization (WHO) estimates that only two of every three patients with MDR TB are diagnosed, three in every four of the diagnosed are treated, and only one of every two of the treated patients are cured, resulting in ~75% of the incident cases persisting in the community or succumbing to their illness. Antibiotic resistance is also an increasing problem in other human pathogens, and transmission of antibiotic resistance from person to person is amplifying the public health threat[2].

Improved surveillance, diagnosis, and treatment are designated priorities by the WHO and the US, European CDCs for addressing the antibiotic resistance challenge[1,3,4]. These measures will rely on an improved understanding of the mechanisms of resistance acquisition in bacteria. The knowledge of genetic mechanisms of antibiotic resistance has formed the basis of several commercial molecular diagnostics for TB that have had remarkable global uptake, despite the fact that they only reliably test for a subset of TB drugs and hence have not yet been able to replace the traditional more costly and slow process of mycobacterial culture and drug susceptibility testing (DST)[1,5–7]. Understanding antibiotic resistance mechanisms and methods that compensate for lost bacterial fitness in the context of antibiotic resistance can also pave the way for the development of companion drugs that restore antibiotic susceptibility[8,9] and can open the possibility of evolutionarily directed therapies that can aid in primary prevention of resistance acquisition[10].

To date, attempts at genome-wide association for antibiotic resistance in MTB have been limited by the relatively low number of isolates phenotypically resistant to antibiotics, and have exclusively relied on phenotypes defined by DST performed at a single critical concentration, likely a result of convenience sampling from clinical isolate archives in clinical mycobacterial laboratories[11–13]. Although such binary DST is currently the standard to guide patient care, MTB critical concentrations lack consistent scientific support and several are based only on consensus[14,15]. The WHO has also declared that "the critical concentration defining resistance is often very close to the minimum inhibitory concentration (MIC) required to achieve anti-mycobacterial activity, increasing the probability of misclassification of susceptibility or resistance and leading to poor reproducibility of DST results"[16]. Although more laborious and expensive, the quantification of the resistance phenotype through MIC testing is considered a major improvement in the current standard for clinical phenotyping of drug resistance[17], and MICs are more appropriate for the assessment of the biological effects of genomic variation in understanding the mechanism of resistance and bacterial fitness. The association of this variation with MICs also promises to refine our molecular prediction of antibiotic resistance for clinical and diagnostic use, as considerable gaps remain in prediction of resistance to first-line drugs like pyrazinamide (PZA), ethambutol (EMB), and second-line drugs including the injectable agents[18,19]. Here, we present a genome-wide association study of 1526 isolates in which MICs were measured for 12 anti-tubercular agents and validate our findings in a globally representative public set of TB genomes with binary DST phenotypic data. We report on 13 non-canonical loci that associate with resistance including two non-coding regions and investigate the role of non-coding variants and interactions for several loci. We additionally measure the heritability of the drug resistance phenotype in MTB and find lower than expected contribution from the known resistance loci.

## Results

**Isolates and resistance phenotype**. Of the total 1526 isolates included in the primary analysis, 76 isolates were excluded because their sequencing data did not meet coverage and mapping criteria (methods). The remaining 1452 isolates originated from 24 different countries, but the majority, 1226, was from Peru. The isolates were each tested against a minimum of four and up to 19 drugs with a median of 12 drugs/isolate (Supplementary Data File 1). Figure 1a provides histograms of the MIC results for isoniazid (INH), PZA, amikacin (AMI), and moxifloxacin (MXF) (complete set of histograms in Supplementary Fig. 1). Overall, 976 isolates were MDR (INH MIC > 0.2 mg per L and rifampicin (RIF) MIC > 1 mg per L) and 438 were pre-extensively drug resistant (XDR) (i.e., additionally resistant to either a fluoroquinolone, MXF, ciprofloxacin or ofloxacin (OFX), or a second-line injectable, SLI, i.e., capreomycin (CAP), kanamycin (KAN), or AMI. A total of 157 isolates were XDR, i.e., MDR and resistant to a fluoroquinolone and a SLI. Despite testing at multiple concentrations close to the critical cutpoint in this sample enriched for MDR, we observed a low rate of intermediate MICs for most first and second line agents with notable exceptions for the drugs EMB, PZA, streptomycin, and ethionamide (ETA) (Fig. 1a and Supplementary Fig. 1).

**Genomic diversity**. We identified 73,778 unique genetic variants in the 1452 genomes. The majority of the variants, 42,871 (58%) occurred in only one of the 1452 isolates (Fig. 1b) and the majority of single-nucleotide substitutions (SNSs) in coding regions were non-synonymous amounting to 36,479 vs 20,541 that were silent. We identified 7178 variants with a frequency of > 0.01 of which 2701 had a frequency of > 0.05. In addition to SNSs we observed an appreciable number of insertions and deletions (indels), with 9% of the observed variants with an allele frequency (AF) > 0.05 being indels.

The isolates' lineage diversity was consistent with their geographic origin with 86% being lineage 4 but diverse within this lineage with 39% of the total being lineage 4.3 (LAM), 31% lineage 4.1 (Haarlem), and 16% representing other L4-sublineages. Of the total, 11% belonged to Lineage 2. There were a total of 43 isolates that belonged to other lineages (L1, L3, and L5). Figure 1c displays the pairwise genetic co-variance between the isolates, and demonstrates that although the majority were lineage 4 there was considerable diversity among the isolates.

**Testing genome-wide association**. Genome-wide association was performed for each drug separately using a gene/non-coding region binary burden score, excluding any loci with burden frequency of <0.01, and correcting for population structure by fitting a linear mixed model. A total of 2791 loci had a burden frequency of ≥0.01. We set the significance threshold at a false discovery rate <0.05 as we planned to perform validation on an independent data set. QQ plots of the resultant p value distribution suggested that the correction for population structure was adequate. This is demonstrated by the adherence of the observed P value distribution to the expected line with the exception of the short tail indicating the significant loci in Supplementary Figs 2 and 3. Twenty known resistance loci (methods) were identified by genome-wide association and for all drugs known loci were associated with the highest effect size and lowest P value of all the significant hits (Supplementary Data File 2). The RNA

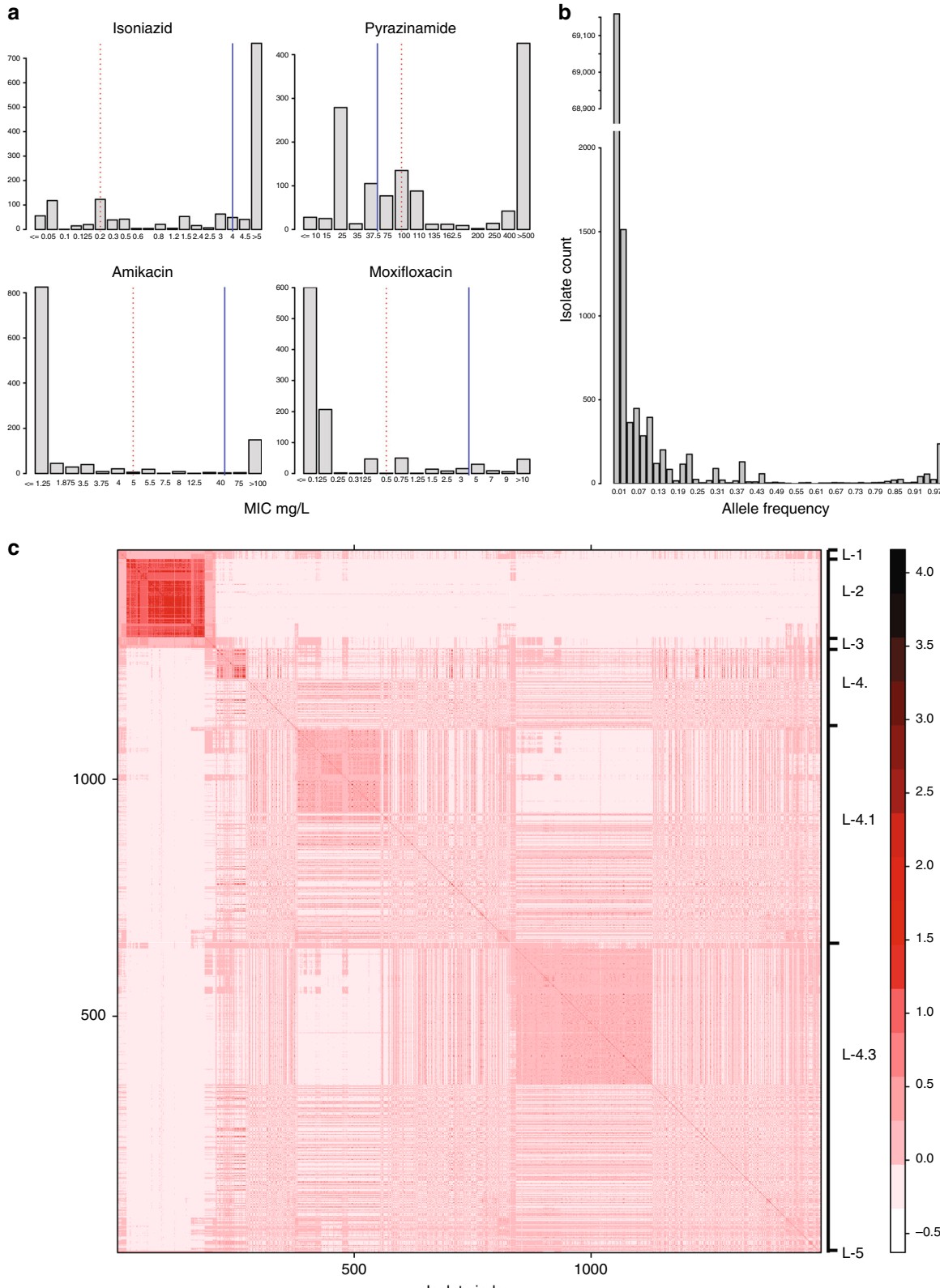

**Fig. 1** Genetic and phenotypic data of 1452 isolates. **a** MIC distributions for four drugs. Dotted red line represents the WHO recommended critical concentration on 7H10 media; blue line represents the lower limit of achievable serum concentration from pharmacodynamic studies (Supplementary Table 2). **b** Allele frequency distribution relative to H37Rv. The isolate count axis interrupted and scaled to accommodate the data range. **c** Heatmap displaying genome-level similarity of the isolates. Similarity assessed as isolate–isolate genetic co-variance. Isolates indexed in lineage order L1, L2, L3, L4 (4., 4.1, 4.3), and L5 as shown. Co-variance ruler displayed on the far right with darker red indicating higher similarity/co-variance. Phylogenetic tree of the same isolates is given in Supplementary Fig. 4. Figure 1a–c can be regenerated using code in Supplementary Data 8 that will refer to data items provided in the Source Data file (compressed source data folder)

**Table 1 Non-canonical regions confirmed in the second validation GWAS and four canonical regions for comparison**

| Locus/site | Drug | AF | Scaled effect size | Scaled SE | P value raw | Drug | AF | OR | SE | P value raw |
|---|---|---|---|---|---|---|---|---|---|---|
| *ubiA (Rv3806c)* | EMB, INH, RIF, PZA, KAN | 0.071 | 0.52 | 0.07 | 1E-13 | EMB, INH, RIF, PZA | 0.066 | 1.25 | 0.10 | 3.6E-03 |
| *Rv3805c - 4267647 T > C (D397G)* | AMI | 0.050 | 2.72 | 0.63 | 4E-06 | AMI | 0.283 | 1.11 | 0.03 | 4.7E-04 |
| *dinG*† | RIF | 0.069 | 1.86 | 0.47 | 2E-05 | AMI, STR | 0.293 | 1.10 | 0.03 | 8.8E-04 |
| *whiB6*^ | CAP, AMI, KAN | 0.037 | 0.59 | 0.15 | 3E-05 | CAP, AMI | 0.069 | 1.16 | 0.06 | 2.7E-03 |
| *RNase J (Rv2752c)* | INH, RIF | 0.042 | 0.77 | 0.20 | 8E-05 | KAN, AMI, RIF, INH | 0.067 | 1.29 | 0.06 | 2.8E-08 |
| *sirA* | ETA | 0.070 | 0.78 | 0.21 | 1E-04 | CAP, KAN, STR | 0.015 | 1.27 | 0.09 | 8.0E-04 |
| *PPE35* | PZA | 0.112 | 0.54 | 0.14 | 1E-04 | EMB | 0.334 | 1.24 | 0.08 | 4.9E-04 |
| *Rv3434c** | KAN, AMI | 0.048 | 1.14 | 0.33 | 1E-04 | KAN, AMI | 0.047 | 1.18 | 0.06 | 2.3E-03 |
| *espK-espL* | RFB, RIF | 0.053 | 1.21 | 0.34 | 2E-04 | AMI, STR | 0.283 | 1.11 | 0.03 | 4.7E-04 |
| *Rv2952*** | EMB | 0.067 | 0.64 | 0.17 | 2E-04 | STR,AMI INH, CAP | 0.203 | 1.29 | 0.08 | 2.6E-05 |
| *ccsA* | KAN | 0.052 | 1.64 | 0.47 | 3E-04 | AMI, CAP | 0.274 | 1.11 | 0.03 | 2.7E-04 |
| *thyX-hsdS.1* | AMI | 0.018 | 0.74 | 0.22 | 3E-04 | STR | 0.016 | 1.32 | 0.13 | 3.9E-03 |
| *kefB (Rv3236c)**** | PZA | 0.137 | 0.80 | 0.24 | 6E-04 | RIF,INH, PZA, EMB STR | 0.235 | 1.60 | 0.16 | 2.0E-06 |
| *inhA* | INH | 0.03 | 1.10 | 0.26 | 2E-05 | – | | | | |
| *Rv1482c-fabG1* | INH | 0.10 | 0.63 | 0.16 | 6E-05 | – | | | | |
| *pncA* | PZA | 0.24 | 1.40 | 0.07 | 4E-89 | – | | | | |
| *pncA-Rv2044c* | PZA | 0.01 | 0.81 | 0.20 | 7E-5 | – | | | | |

All drugs to which they were found to be associated are listed. The first drug listed was the drug found to be most significantly associated and for which the GWAS results are listed in the subsequent columns. Left half of the table represents the test GWAS results ($n = 1452$) and the rightward half lists results of the validation ($n = 792$). Four known/canonical resistance loci are also listed along with their respective allele frequency, effect size, and P value for comparison. Full results detailed in Supplementary Data Files 2–3. Drug abbreviations detailed in Supplementary Table 2. AF: allele frequency. OR: odds ratio. SE standard error. Scaled effect size and SE are on the logMIC scale. *transmembrane protein, **integral membrane transport protein, ***probable methyl transferase and membrane protein, ^transcriptional regulator, †structure-specific helicase

polymerase β-subunit gene *rpoB* was the most significant hit across all drugs with a RIF logMIC increase of 3.24 log(mg per L) and Wald $P$ value of $<10^{-187}$. Of the known locus–drug associations detected, the smallest effect size was measured for the *embA–embC* intergenic region, an EMB logMIC increase of 0.45 at a Wald $P$ value of $1 \times 10^{-7}$. Notably, we did not identify a significant association between the compensatory gene *rpoA* and RIF resistance, the *embA* and *embC* genes and EMB resistance and between *gyrB* and MXF resistance. Given stepwise and co-linear development of antibiotic resistance in MTB and the prevalence of MDR in our sample, most of the known resistance loci were identified to be associated with more than one antibiotic, but in each case the known causative locus was the most significantly associated with its respective drug (Table 1, Supplementary Data File 2). We implicated several promoter/intergenic regions surrounding known genes including not only the *Rv1482c-fabG1* and the *eis-Rv2417c* intergenic regions that are currently used in one or more commercial diagnostics[6,20], but also the regions upstream of *embAB* (*embA–embC*), *pncA* (*pncA-Rv2044c*), and *ahpC* (*oxyR'–ahpC*). The known compensatory gene *rpoC* was strongly associated with resistance to both RIF and rifabutin. We also identified the *rpsA* gene to be associated with PZA resistance with an effect size and $P$ value lower that of variants in the intergenic region containing the *pncA* promoter (0.55 logMIC increase and $2 \times 10^{-4}$ vs 0.81 and $7 \times 10^{-5}$, respectively, Supplementary Data File 2).

We identified 50 non-canonical loci to be associated with resistance to one or more antibiotics (Supplementary Data File 2). Sixteen loci were associated with resistance to more than one drug. Two such loci were associated with resistance to all three SLI agents, the gene encoding the transcriptional regulator WhiB6, the cytochrome P450 oxidoreductase encoding *fprA* gene (logMIC change and Wald $P$ value: 0.59 and $1 \times 10^{-4}$, 1.37 and $1 \times 10^{-6}$, respectively). CcsA a gene in the

cytochrome P450 maturation pathway was also associated with SLI resistance (KAN logMIC change 1.64 and Wald $P$ value $2 \times 10^{-4}$—Table 1 and Supplementary Data File 2) with an effect size among the top 10 measured for the non-canonical loci. The most significantly associated non-canonical locus was the gene *ubiA* (Rv3806c) with the drug EMB (logMIC 0.52 and Wald $P$ value $1 \times 10^{-13}$). The locus *Rv3083* which encodes the gene *mymA*, an alternative monooxygenase to *ethA*[21], was associated with resistance to ETA and two other drugs and was among the 10 most significant non-canonical hits (ETA logMIC 0.60 and Wald $P$ value $1 \times 10^{-4}$). Twelve intergenic regions were found to be associated with resistance including the intergenic regions *thyX-hsdS.1* and *glnE-glnA2*, as well as regions adjacent to type VII secretion system related genes like *espK-espL* (Table 1 and Supplementary Data File 2). The secondary genome-wide association performed at the site level identified associations of individual substitutions (SNS) or indels within the loci associated in the primary analysis (Supplementary Data File 2). In addition, four SNSs in other non-canonical loci: L111M in *Rv3327*, D397G in gene *aftB*, and 640954AG in the intergenic regions *Rv0550c-fadD8* were associated with resistance (Supplementary Data File 3). No non-canonical associations were found for the drug linezolid.

**Validation in an independent data set**. The 50 non-canonical associations were tested in an independent set of globally representative MTB isolates with public sequence and drug resistance data. The validation set showed a higher level of genetic diversity with 44.3% of the 792 isolates belonging to lineage 2, 40.3% belonging to lineage 4 (15% 4.1 sublineages, 8% 4.3 sublineages) and a higher representation of other lineages: 5% L1, 4% L3, 3% L6/BOV/AFR. The proportion of isolates that were MDR in the validation set was 35% (278 isolates). Second-line drug resistance

phenotypes were available for 25–57% of the isolates (Supplementary Data File 4) and 29 isolates were XDR. Of the 50 loci identified above, six could not be validated as there was no appreciable variation observed in the set of 792 isolates (AF < 0.01). Twenty seven other loci were tested but had an AF < 0.05 and were not significantly associated, these included the loci *mymA* and *fprA*. Of the remaining 17 loci, 12 were validated to be associated with resistance to one or more drugs. These included *whiB6, ccsA, ubiA*, a metal beta-lactamase *Rv2752c*, and two intergenic regions including *thyX-hsdS.1* (Table 1). In the site level analysis the D397G mutation in the gene Rv3805c (*aftB*) was validated as significantly associated with resistance. The strength of association for several of the non-canonical loci was comparable to some canonical genes, but the allele or burden frequency was lower for most of them. For example the effect of *ubiA* mutations on the EMB MIC was measured to be 0.52 logMIC increase, similar in magnitude to the effect of variants in the *Rv1482c-fabG1* intergenic region on INH MIC (0.63 logMIC increase) as was the effect of *whiB6* mutations on SLI MICs (ranging between 0.56 and 0.60 logMIC increase). The respective allele frequencies were 0.07 for *ubiA*, 0.03–0.04 for *whiB6* and 0.10 for the *Rv1482c-fabG1* intergenic region. The allele frequency i.e. the frequency of the minor variant within our sample, was < 10% (Table 1, Supplementary Data File 2) in all but two validated loci.

All of the validated regions were found to have variants in two or more of the major TB lineages, and all but three of the coding loci harbored nonsense or frameshift variants in one or more isolates (Supplementary Data File 5, Supplementary Fig. 3). In a formal test for phylogenetic convergence[22], *ubiA, whiB6, Rv2752c, PPE35, Rv3236c,* and *thyX-hsdS.1* displayed significant homoplasy at a permutation $P$ value < 0.005 (Supplementary Data File 6). The distribution of variants varied by locus; *ubiA, whiB6, Rv2752c* and *PPE35* all displayed considerable diversity of variants that were closely spaced in one or more segments of the gene, in a pattern similar to that observed in known resistance genes (Supplementary Fig. 3). For the intergenic hits, we observed a concentration of variants around the predicted transcriptional start site in both cases (Supplementary Fig. 3).

**Resistance phenotype heritability**. We examined the proportion of variance in the resistance phenotype explained (PVE) by all of the observed genetic variation for each drug (Table 2). The PVE varied by drug, ranging from $0.64 \pm 0.06$ (standard error) for MXF and $0.66 \pm 0.04$ for PZA at the lower end to $0.84 \pm 0.02$ for RIF and $0.88 \pm 0.02$ for AMI at the higher end. We measured the PVE for the known antibiotic resistance genes, and that for non-

canonical genes captured in this study. The proportion explained by the known genes was relatively low and at most $0.24 \pm 0.08$ (27% of the total PVE) for AMI. The proportion explained by the non-canonical genes was even lower but on par with PVE of known drug resistance loci for PZA and ETA albeit with large error margins (Table 2).

**Interactions with canonical resistance regions**. We sought to determine whether there are detectable interactions between specific resistance sites, genes, and the genetic lineage. We hypothesized that because antibiotic resistance arises as a result of strong positive selection in MTB and several sites have large effects on the phenotype that the MIC distributions observed for such resistance mutations would not vary appreciably across different lineages. We focused on lineage 4 and lineage 2, as they were well represented in our sample. Examining the six mutations: *katG* S315T, *rpoB* S450L, *rpoB* D435V, *embB* M306V, *inhA* -15, *pncA* H51R for INH, RIF, RFB, EMB, ETA, and PZA, respectively, we found the MIC distributions to not to be appreciably different for five of the six mutation–drug pairs (Wilcoxon $P$ value > 0.2). We did associate the mutation *rpoB* D435V with higher median rifabutin MICs among lineage 2 isolates than among lineage 4 (median 0.375 mg per L vs. ≤ 0.125 Wilcoxon $P$ value $6 \times 10^{-4}$). Examining interactions between specific pairs of mutations and genes, we first tested if the acquisition of additional resistance mutations causative of resistance to other drugs is associated with increases in MIC. We focused on the first line drug EMB for which the most significant non-canonical hit *ubiA* was identified as well as the second line aminoglycoside KAN. We examined the loci *embB, embA, embC*, and *ubiA* for EMB and *rrs* for KAN. We found EMB MIC levels to be higher among isolates with both an A1401G *rrs* variant and an M306V *embB* variant, as compared with those with M306V *embB* and without the *rrs* variant (Wilcoxon $P$ value 0.005 median > 15 (IQR 12.5–15) vs > 15 (IQR > 15– > 15). *UbiA* and *embA* variants were also more common among isolates with both M306V *embB* and A1401G *rrs* compared with isolates harboring only the former (*embA* OR 13.8 (95% CI 6.1–33.6, Fisher $P$ value < $10^{-12}$), *ubiA* OR 6.7 (95% CI 3.3–13.8, Fisher $P$ value < $10^{-8}$)). After excluding isolates with *embA* and *ubiA* variants, isolates with both the *rrs* and *embB* variant still tended to have a higher MIC but the $P$ value decreased to 0.05. Isolates with both *embA* and *embB* variants were more likely to have a higher EMB MIC (median > 15, IQR 7.5– > 15) than those with only *embB* variants (median > 15, IQR 3.5 to > 15, Wilcoxon $P$ value $4 \times 10^{-4}$). The co-occurrence of *embB* and *ubiA* variants was also associated with elevations in the EMB MIC relative to *embB* variants alone:

**Table 2 PVE for each drug attributable to subsets of genetic variations**

| Drug | All PVE | All SE | woDR PVE | woDR SE | woDRwoNC PVE | woDRwoNC SE | DR-related PVE | NC-related PVE |
|---|---|---|---|---|---|---|---|---|
| INH | 0.809 | 0.020 | 0.732 | 0.032 | 0.723 | 0.029 | 0.08 | 0.01 |
| RIF | 0.838 | 0.017 | 0.701 | 0.034 | 0.692 | 0.033 | 0.14 | 0.01 |
| RFB | 0.833 | 0.023 | 0.722 | 0.041 | 0.693 | 0.042 | 0.11 | 0.03 |
| EMB | 0.748 | 0.027 | 0.674 | 0.036 | 0.665 | 0.035 | 0.07 | 0.01 |
| PZA | 0.659 | 0.038 | 0.634 | 0.044 | 0.602 | 0.043 | 0.03 | 0.03 |
| KAN | 0.833 | 0.022 | 0.671 | 0.040 | 0.658 | 0.041 | 0.16 | 0.01 |
| AMI | 0.879 | 0.019 | 0.639 | 0.057 | 0.640 | 0.055 | 0.24 | <0.01 |
| CAP | 0.743 | 0.030 | 0.690 | 0.038 | 0.666 | 0.038 | 0.05 | 0.02 |
| ETA | 0.701 | 0.034 | 0.689 | 0.038 | 0.675 | 0.038 | 0.01 | 0.01 |
| STR | 0.710 | 0.033 | 0.604 | 0.047 | 0.601 | 0.045 | 0.11 | <0.01 |
| MXF | 0.643 | 0.058 | 0.494 | 0.083 | 0.456 | 0.081 | 0.15 | 0.04 |

PVE all measurable genetic variation given along with PVE related to variation excluding known drug resistance regions and non-canonical regions associated in this study (*n* = 1452 isolates). Drug abbreviations detailed in Supplementary Table 2. PVE: proportion of variance explained. DR: drug resistance regions as detailed in the methods. NC: Non-canonical regions specified in Table 1. wo: without. SE: standard error

median of > 15, IQR 7.5– > 15 to a median of > 15, IQR 12.5 to– > 15, Wilcoxon *P* value $6 \times 10^{-4}$. On the other hand, variants in embC were not more likely to co-occur with embB variants, Fisher *P* value 0.6; and there was no difference in EMB MIC between isolates with both *embC* and *embB* variants vs. those with only *embB* variants, Wilcoxon *P* value 0.5.

**Resistance effect of promoter vs. gene body variants.** Given the number of intergenic regions found to be associated with resistance we tested the hypothesis that intergenic variants have smaller effects on drug MIC compared with gene body mutations for the three genes and the promoter-containing upstream intergenic region that were independently associated with resistance in the GWAS; namely *inhA, pncA,* and *embB* and their upstream intergenic regions, respectively. We focused on the codon and promoter site with the largest allele frequency in each case. Isolates not infrequently had both a gene body and a promoter mutation: 12% of isolates with *embB* promoter mutations also had an *embB* codon 306V, and 18% of isolates with an *inhA* promoter mutation also had a mutation at *inhA* codon 21. None of these variants were phylogenetically restricted (Supplementary Table 1). No isolates had both a *pncA* promoter mutation and a *pncA* mutation at codon 51. Figure 2 shows the marginal MIC values for each site pair and drug. Variants in promoter regions consistently showed lower MICs than gene body mutations, although in most cases both medians were above the clinical cutoff (*P* values 0.03, 0.002, 0.01, 0.009 for the drugs INH, ETA, PZA, and EMB, respectively). This findings were also supported

by the relative magnitude of the GWAS regression coefficients at the locus level for each drug (Table 1) with the notable exception of ETA (Supplementary Data File 2). We also tested the possibility that genes that harbor promoter variants associated with resistance were more likely to be essential genes than genes that exclusively harbor variants in the gene body in association with resistance. Of the latter, 7/11 were essential, whereas only 3/7 genes with promoter resistance variants were essential, suggesting that gene essentiality is limited in assessing the functional impact of a variant.

**Discussion**
Here, we examine 1452 clinical MTB isolates, enriched for phenotypic resistance, and quantify their antibiotic resistance phenotype using the MIC method. Our GWAS results using this quantitative phenotype are notable for the capture of several non-coding genetic regions. In aggregate, > 20% of the loci associated with antibiotic resistance were intergenic regions. This stands in contrast to the relatively low proportion of the MTB genome annotated to be non-coding, 10.5% by length for the H37Rv reference. Although only a subset of these regions are known promoter regions, their association with antibiotic resistance, and the concentration of the variants around predicted transcriptional start sites raises the possibility that the non-canonical regions may also play a role in gene regulation. Canonically, antibiotic resistance is caused by protein-modifying mutations in drug targets or in pro-drug to drug-converting enzymes in MTB. Also, to date, commercial-based assays for detecting antibiotic

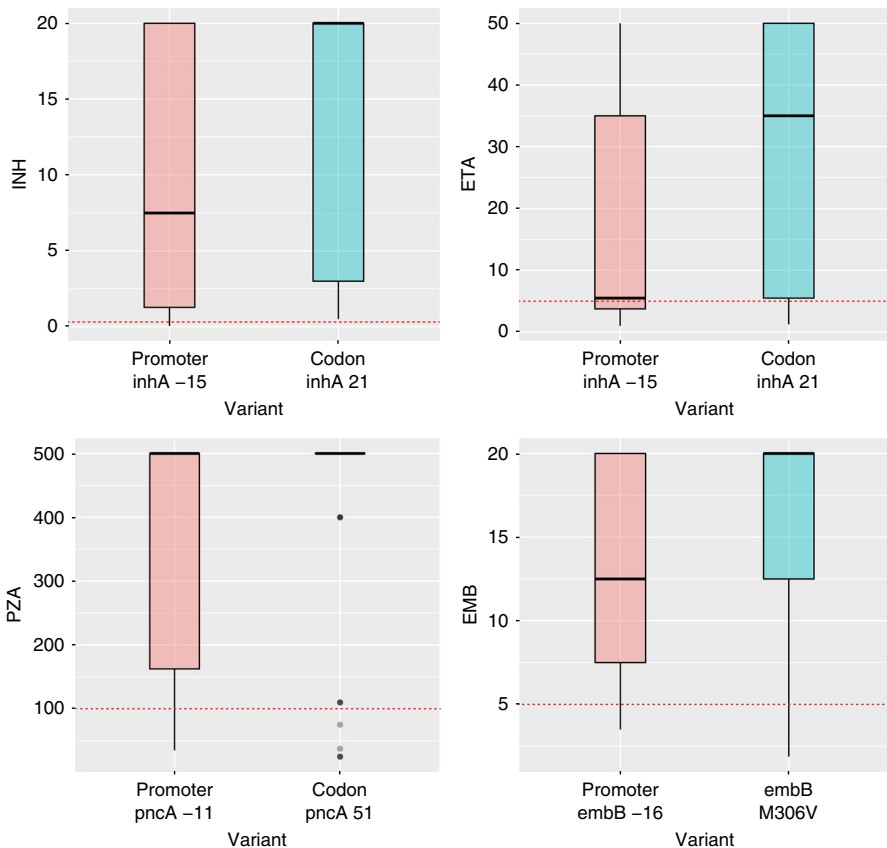

**Fig. 2** Promoter vs gene body mutations and their effect on MIC (*y* axis in mg per L) for four drugs (*n* = 1452). Wilcoxon rank sum test comparing each pair was significant at *p* < 0.01. For reference critical resistance testing concentration is 0.2 mg per L for INH, 5 mg per L for ETA, 100 mg per L for PZA, 5 mg per mL for EMB and indicated by the horizontal dotted line. Figure 2 can be regenerated using code in Supplementary Data 8 that will refer to data items provided in the Source Data file (compressed source data folder)

resistance in TB have largely focused on gene based variants, with the notable exception of the *inhA* and *eis* promoters. We find that isolates harboring mutations in promoter regions tend on average to have lower drug MICs than those isolates with a corresponding non-synonymous gene body variant, and although these tend to exceed the critical cutpoint in both cases, if the MICs are close enough to the cutpoint the isolates may be treatable in some cases with higher doses of drug or a more potent drug from the same class[23,24]. This highlights the importance of understanding the underlying genetic cause of resistance and personalizing therapy based on this, but definitely requires further investigation including potentially clinical trials exploring the efficacy of higher dose antibiotic therapy in patients with such isolates.

We identify and validate 12 genetic regions and one SNS as associated with resistance in MTB. Although these loci have, to date, not been used to predict or diagnose antibiotic resistance in patients with TB[18,19,25], several have been recently associated with resistance either in vitro or in other genome-wide association studies performed on binary resistance read outs (Supplementary Data File 7)[11–13]. We summarize these by drug or class, detailing the full results in Table 1 and Supplementary Data Files 2–3 and 7.

The gene with the most significant *p* value in the primary GWAS was *ubiA*. This locus is validated further by the results of two prior GWAS studies[13,26], and mutations introduced at *ubiA* codon 237 were shown to increase gene function and elevate decaprenylmonophosphoryl-β--ribose or arabinose (DPA) levels[27,28]. DPA is the donor substrate for arabinosyltransferases that include EmbB, the main target of the drug EMB, and increases in DPA levels likely result in competitive inhibition of the EMB drug effect measurable as an increase in the EMB MIC. The downstream gene *aftB* that encodes an enzyme catalyzing the final step in arabinoglycan arbinan biosynthesis was also found to have a SNS significantly associated with resistance in our study. The association was not with EMB but rather with the drug AMI, as most AMI resistance isolates are also EMB resistant we suspect this mutation to be compensatory to EMB resistance rather than resistance causing, reinforcing *aftB* to be a potentially valuable drug target as has been previously suggested[29,30].

The gene Rv2752c encodes a bifunctional beta-lactamase /ribonuclease[31,32], and was found to be associated with resistance in one prior survey[26]. We found this gene to be associated with resistance to either INH or RIF, with an effect size comparable to that of *inhA* promoter mutations on INH resistance, but with an allele frequency that was half of that of *inhA* promoter mutations (at 0.05). The integral membrane transport protein *KefB* (Rv3236c) is a $K^+/H^+$ antiporter that releases $K^+$ to the phagosomal space and prevents its acidification. We found variants in the encoding gene to associate with resistance most strongly with PZA resistance, which is compelling given the known modulating effect of the medium's pH on PZA's drug activity[33].

We found several non-canonical associations most strongly with the AG class of anti-tuberculosis drugs. These include the transcriptional regulator whiB6 that is known to activate expression of the DosR regulon, and controls aerobic and anaerobic metabolism and virulence among other pathways[34]. Previous work has implicated another *whiB*-like transcriptional regulator, whiB7, in resistance to AGs[35] and whiB6 and the upstream intergenic region were previously associated with resistance in a prior GWAS albeit to non-AG agents[12]. The cytochrome-c maturation gene *ccsA* encodes an integral membrane protein that binds heme in the cytoplasm and exports it to the extracellular domain of *ccsB* that in-tail primes it for covalent attachment to apocytochrome *c*. Deficient cytochrome-c oxidase activity is tolerated in MTB owing to the flexibility of its electron

transfer chain[36], it is plausible that this may incur a fitness advantage by slowing growth under drug pressure. *CcsA* and *katG* variants commonly co-occurred in one genomic analysis of 288 isolates from China and the pairs were found to be significantly associated with resistance[37].

The gene *mymA*, an alternative monoxygenase to *ethA*[21], which encodes an enzyme known to activate the pro-drug ETA, was associated with an increase in ETA MIC. In vitro, *mymA* deletion mutants were previously found to be resistant to ETA, and double *mymA* and *ethA* knock out mutants had even higher ETA MICs that the individual mutants[21]. We were not able to validate *mymA* in the independent data set against the binary ETA resistance phenotype, possibly owing to limited statistical power as only 116 isolates in the validation data set were ETA resistant, and *mymA* variants are more rare occurring in < 5% of the isolates in the test set. It is also possible that *mymA* mutations increase ETA MIC to a smaller extent in clinical isolates making the GWAS against binary phenotypes less sensitive. The diagnostic utility of *mymA* mutations for improving the prediction of ETA thus requires more study.

Mutations in the intergenic region upstream of *thyX* (*thyX-hsdS.1*) have been shown to modulate *thyX* expression and have been associated with resistance in two MTB GWAS[13,26]. Given that *thyX* is involved in folate metabolism, mutations in these regions may be causative of or compensatory for PAS resistance[13]. It is notable that the association we measured was with respect to other drugs, INH, KAN, and AMI, as we did not have a sufficient number of isolates tested for PAS resistance. This likely resulted from drug-drug resistance collinearity and emphasizes the need to carefully interpret novel GWAS results in MDR-bacteria.

The measurement and genome-wide association with MICs allowed us to quantify, for the first time, the proportion of the TB resistance phenotype that is explained by bacterial genetic variation. We estimate that 64–88% of the MIC variance to be explained by genetic effects, with standard errors ranging from 2 to 6%. The remaining proportion may be explained by other factors such as genetic interactions, mutation heterogeneity, or environmental or other testing related factors that result in MIC level variability. It is notable that we found the known resistance loci to explain a relatively low amount of the total variation ranging as low as 0.01 for ETA to 0.24 for AMI. The gap between total PVE and that attributable to known drug resistance loci, is not completely explained by the presence of the non-canonical genetic loci as these explained an even lower proportion than known drug resistance loci, likely related to their low mutation frequency. This gap may be better explained by lineage or gene–gene interactions. We did examine a set of specific interactions between six canonical resistance mutations and genetic lineage (lineage 4 vs 2), and between variants in the loci *embABC* and *ubiA* one of the non-canonical candidates. Although for several of the loci we examined that individually had large effects, like *katG* S315T and *rpoB* S450L, we could not detect an interaction with the genetic background, in the case of *ubiA*, *embA*, and *rpoB* D435V we found evidence for the presence of at least additive interactions on the drug MIC[38]. It thus seems likely that such interactions exist for other mutations, and may be widespread in bacterial genomes, especially between variants with smaller effects. These conclusions are consistent with prior evidence from allelic exchange experiments[39].

In this study, we demonstrate the utility of genome-wide association for examining bacterial phenotypes relevant to infectious disease. Our study was not without limitations. To achieve sufficient statistical power to detect associations between genetic variation and clinically relevant resistance phenotypes we oversampled isolates that had higher levels of drug resistance. It is

possible that this over-representation of high level resistance enriched for a subset of genetic variants and make it less likely to capture variants that are rarer or have smaller effects. Given the recognized step-wise acquisition of resistance in MTB[40], it is very challenging to determine accurately which drug resistance is in fact associated with a particular gene or genetic region. For example, resistance to any of the second line agents, fluoroquinolones like MXF, SLI's like AMI, or to first-line agents like PZA and EMB, nearly always co-exists with resistance to INH and RIF, and it is thus not possible to perform association conditioning on the absence of resistance to those agents. This also significantly limits our ability to assess for mutations that can result in cross-resistance between drug classes; however, our results do support recognized associations between genetic loci and drug members of the same class such as *inhA* promoter variants with INH and ETH resistance, *rrs* variants and AMI, KAN, or CAP resistance and *rpoB* variants and RIF or RFB resistance (Supplementary Data Files 2 and 3). Further, the performance of linear mixed models for performing GWAS in bacteria has not been systematically studied, although applied recently to MTB and other bacteria with demonstrated success[13,41–43]. We acknowledge that we cannot be certain that these models adequately control for population structure in clonal bacteria and because of this we performed validation in an independent data set with a different lineage distribution. We also provide the lineage breakdown of variants in our hit loci that in each case demonstrated evidence for convergent evolution[11]. We also demonstrate the power of using a binary gene-burden score for bacterial GWAS, as this decreased the number of necessary tests relative to GWAS of individual sites and allowed the incorporation of rare genetic variants that appear to be important for drug resistance in MTB[44]. This approach is however, reliant on the accuracy of the available genomic annotation for MTB, and is most sensitive for capturing genes under diversifying selection, i.e., where multiple different genetic mutations may contribute to a functional genetic change, and entirely ignores synonymous variation as potentially contributing to the phenotype. More refined measures of gene burden in bacteria, for example, measures that incorporate protein structural data, are worth investigating systematically in the future.

In summary, with the increasing availability of genomic data, powered by the formation of TB genomic data consortia[45], our ability to identify more-rare variants with smaller effects on resistance will increase. Our improved understanding of the genetic mechanisms of resistance in MTB can perhaps lead to more-targeted drug development efforts, but more imminently will allow for improved diagnosis and surveillance given the increased uptake of genomic technologies in public health laboratories in high income countries[46]. Improvement in portable sequencing technology[47] and decreased cost of sequencing is promising to facilitate adoption in settings with lower resources where TB is most prevalent. However, even if sequencing technology is available, our results suggest that genomic data interpretation will likely necessitate the use of statistical models or machine learning[18,44,48] given the number of genetic loci associated with resistance and the likely contribution of gene–gene interactions, especially if a quantitative prediction of the drug MIC is desirable[17]. The portability of the potential benefits of these advances to areas of the world where TB is most prevalent and will require continued efforts in open sharing of data and analysis tools[49].

## Methods

**Sample collection**. MTB sputum-based culture isolates were selected from (1) a Peruvian patient archive of culture isolates enriched for resistance based on prior targeted resistance gene sequencing and binary DST phenotype[18] ($n = 496$), or (2)

sampled from a longitudinal cohort of patients with Tuberculosis from Lima Peru[50] enriched for multidrug resistance based on prior binary DST ($n = 568$). These 1064 isolates had phenotypic resistance testing by MIC for 12 drugs repeated (see below) at the National Jewish Hospital (NJH) Denver, CO, and underwent whole-genome sequencing. Data from these isolates were pooled with data from two additional samples: a convenience sample from three national or supranational reference laboratories selected based on the availability of MIC data: the Institute for Tropical Medicine–Antwerp, Belgium, the Massachusetts State TB Reference Laboratory–Boston, MA, and the National Institute for Public Health and the Environment–Bilthoven, Netherlands ($n = 411$) and a sample of 83 pan-susceptible isolates from the Peruvian TB cohort[50] added to increase the representation of sensitive isolates. This study protocol was reviewed by the Harvard Medical School Institutional Review Board and was designated not to constitute research on human subjects as it included only de-identified microbiological samples.

**Culture and drug resistance/MIC testing**. Lowenstein–Jensen (LJ) culture was performed from sputum specimens using standard NALC–NaOH decontamination. Prior to DNA extraction and sequencing most cultures had been cryopreserved as follows: Inside a biosafety container, all colonies of each culture were extracted from the LJ slants and dissolved in 7H9 broth with 20% glycerol to reach a bacterial suspension similar or higher than McFarland 5. Then, the bacterial suspension was aliquoted in volumes of 0.3–0.5 mL and stored overnight at 4 °C to ensure the glycerol uptake of the cells. Then, all tubes were placed into the − 80 °C freezer for long term storage.

All isolates, except the 83 pan-susceptible isolates described above, underwent MIC testing. Testing for the 1064 isolates at NJH was performed for 11 anti-TB drugs on 7H10 media using agar proportion and for PZA in MGIT 960 in a staged fashion. Isolates were first tested at three low concentrations that include the WHO recommended critical concentration. If the isolate was resistant at the critical concentration then testing at six higher concentrations was additionally performed. The testing concentrations deviated from the traditional doubling to better detect intermediate level MICs that are close to the clinical critical concentration and within theoretically achievable levels in patient sera based on available pharmacodynamics data[51]. The concentrations are detailed in Supplementary Table 2. Culture, MIC, and DST testing at the other laboratories is outlined in Supplementary Table 3. Testing methods and concentrations are also listed for each isolate in Supplementary Data File 1.

**MIC quality control procedures at the NJH consisted of the following two measures**. (1) The repeat testing, with every batch, of two external control MTB strains, one resistant to all drugs except MXF and linezolid and the other susceptible to all drugs. Testing of these two reference strains was repeated using the same method (indirect proportions on 7H10 agar) with each batch of ~ 30 clinical isolates to confirm the reference MICs replicate at the exact level. If either or both of the reference strains failed to replicate the expected MIC for one or more drugs, the whole batch of isolates was re-tested. Of the 33 batches and replicate reference MTB strain tests conducted during the 2.5 years of testing, only one failed to replicate and was repeated. The reference MICs were reproduced upon the repeat testing.

(2) Internal controls: every clinical isolate was tested on an agar plate split into four quadrants, three quadrants contained increasing concentrations of the drug and the fourth was a control quadrant free of drug. If the isolate failed to grow in the control quadrant (at least 50 colonies), the isolate was re-tested (i.e., on a new plate) given the concern for inadequate innoculum. The culture plates were also monitored for contamination. If any contamination was observed the isolate was also re-tested. Over 29/1091 isolates were re-tested because of there was either no-growth or contamination. Two isolates of the 29 could be grown and MICs for these isolates were run twice repeated. In both cases the MICs were replicated to within one MIC dilution.

**DNA extraction and whole-genome sequencing**. DNA from sputum samples of TB patients was extracted from cryopreserved cultures. Each isolate was thawed and subcultured on LJ and a big loop of colonies were lysed with lysozyme and proteinase K to obtain DNA using CTAB/Chloroform extraction and ethanol precipitation. DNA was sheared into ~ 250 bp fragments using a Covaris sonicator (Covaris,Inc.), and prepared using the TruSeq Whole-Genome Sequencing DNA sample preparation kit (Illumina, Inc.). Samples were sequenced on an Illumina HiSeq 2500 sequencer. Paired-end reads of length 125 bp were collected. Base-calling was performed using HCS 2.2.58 and RTA 1.18.64 software (Illumina, Inc.)

**Definition of known and non-canonical drug resistance loci**. We define the MTB known resistance loci as the following genes *katG*, *inhA*, and its promoter, *ahpC* promoter, *kasA*, *rpoB*, *embA*, *embB*, *embC*, and *embA–embC* intergenic region, *ethA*, *gyrA*, *gyrB*, *rrs*, *rpsL*, *gid*, *pncA*, and its promoter, *tlyA*, *thyA*, *rpsA* promoter, and the compensatory genes *rpoC*, *rpoA* based on prior published work[6,18,20,25,52–54] and the use of many of these regions in commercial molecular diagnostics for MTB. We define loci other than those listed above as non-canonical loci if they were found to be significantly associated in the GWAS.

**Variant calling and phylogeny construction**. Genome coverage was assessed using SAMtools 0.1.18[55] and FastQC[56] and read mapping taxonomy was assessed using Kraken[57]. We aligned the Illumina reads to the reference MTB isolate H37Rv NC_000962.3 using Stampy 1.0.23[58] and variants were called by Platypus 0.5.2[59] using default parameters. Strains that failed sequencing at a coverage of < 95% at ≥ × 10 of the known drug resistance regions, or that had a mapping percentage of < 90% to *M. tuberculosis* complex were excluded. Genomic regions not covered at ≥ × 10 in at least 95% of the remaining isolates were filtered out from the analysis, i.e., no attempt at association with variants in those regions was made. In the remaining regions, variants were further filtered if they had a quality of < 15, purity of < 0.4 or did not meet the PASS filter designation by Platypus. We used the purity threshold of 0.4 as in a previous comparison with a lower threshold (of 0.1) there was no significant improvement in sensitivity over specificity[18]. We also excluded any indels > 3 bp in size or large sequence polymorphisms. Further quality control was performed after genome-wide association when associated PE/ PPE gene and indels were visualized and manually inspected using IGV v2.4.9[60]. TB genetic lineage was called using the Coll et al.[61]. SNP barcode and confirmed by constructing a Neighbor joining (NJ) phylogeny using MEGA-5[62], excluding known resistance genes and potentially repetitive regions[63] and including lineage representative MTB isolates from Sekizuka et al.[64].

**Phenotype**. The MIC data were recorded as an interval indicating the last highest concentration tested where growth was seen and the MIC itself. Because critical concentrations on LJ media (for isolates tested at ITM) are in general higher than those on 7H10, the MIC intervals were normalized to allow for comparability by dividing by the critical concentration for each drug as defined by the WHO[65]. The interval midpoints were computed and converted to ranks as has been previously suggested for genotypic association with MIC data[22]; ties were assigned an average rank. A sensitivity analysis was performed to confirm that the results are not sensitive to the rank transformation of the phenotype, by comparing the region hits obtained in a parallel GWAS analysis using the natural log transformed phenotype instead of the rank transform.

**Genotype, GWAS, and control for population structure**. Association analysis was performed at the gene/non-coding region level using a binary gene burden score that was set at one if any non-synonymous SNS or indel (insertion or deletion) was observed in a gene, or any SNS or indel was observed in a non-coding region, and zero otherwise. We excluded known lineage markers in drug resistance genes from the burden score calculation[18]. Association was also performed at the site level in a secondary analysis excluding synonymous variants. Any gene/region or SNS with a minor allele frequency (MAF) of <0.01 was not tested. We controlled for population structure by computing a genetic relatedness matrix (GRM) that measures genetic similarity as the co-variance between the individual isolate genetic variant vectors. For the GRM computation we included all synonymous and non-synonymous SNSs and indels but excluding variants in known drug resistance loci and variants occurring at a MAF of < 0.01 using the software package GEMMA[66]. Genome-wide association was performed using a linear mixed model with the phenotype as the rank-transformed MICs also using GEMMA. Regions with a false discovery rate <0.05 were selected for validation. We verified control for population structure with QQ plots using the qqman package in R v3.2.3. As the regression was performed on rank-transformed MIC values, we scaled the resulting effect size back to the MIC scale by first performing a linear regression between the natural logMIC values and their rank transform and then using the resulting slopes as a scaling factor. LogMIC change in units of log(mg per L) are reported throughout. In a parallel analysis we ran a test for phylogenetic convergence[11] using the MEGA-5 NJ tree (Supplementary Fig. 4) and the treeWAS R package[22] utilizing the simultaneous score and a permutation $P$ value threshold of <0.005 to assess significance. For comparisons of isolate proportions harboring a specific genotype we used the Fisher exact test, and compared MIC distributions using the Wilcoxon rank sum test. For these latter two, the $P$ value threshold was <0.05.

**Validation**. We validated the genomic regions identified above in an independent public data set with binary phenotype data. The validation data set consisted of a convenience sample of 792 MTB isolates obtained by pooling data from the ReSeqTB knowledge base (https://platform.reseqtb.org/)[45] with additional MTB whole-genome sequences and phenotype data curated manually from two additional references[26,67] (Supplementary Data File 4). We did not select isolates for the validation set based on lineage or drug resistance profiles. Association analysis was performed using a linear mixed model approach as was outlined above for the test data and using a GRM for population structure correction. A locus was considered validated if it had a Wald $p$ value of < 0.005.

**PVE**. We computed the PVE as the proportion of total phenotypic variance explained by the genetic relatedness between the isolates, using the restricted maximum likelihood approach as implemented in GEMMA, as a measure of heritability. We computed the PVE attributable to known drug resistance regions by recomputing the GRM after removing all variation (synonymous, non-synonymous, and indels) in the known resistance loci. Similarly, we

computed the PVE attributable to all other loci validated to be significantly associated with resistance in this study, as PVE attributable to the non-canonical loci. Given the phenotypes were coded as ranks of the MIC distribution, we performed a sensitivity analysis to confirm that rank transformation did not affect our PVE measurements. In this sensitivity analysis we dichotomized the MICs using the WHO-established critical concentration as the threshold, and recomputed the PVE on the liability scale. The PVEs changed by <10% for all drugs in the sensitivity analysis.

## Data availability
All data used in this study are available in the supplementary material or deposited on NCBI with accession numbers detailed in Supplementary Data 1. The source data and code underlying manuscript figures as well as the results described are provided in the Source Data file (compressed source code folder) and in Supplementary Data 8 (R-code described in the Code Availability section above). Figure 1a–c, Fig. 2, Tables 1, and 2 can be regenerated using code in Supplementary Data 8 that will refer to data items provided in the Source Data file. Any other information is available upon reasonable request from the corresponding author

## Code availability
R-code for constructing figures and processing GWAS output can be accessed in Supplementary Data 8 (source code), and additional bash code for running the GWAS and heritability calculations is available within the Source Data file (compressed source data folder).

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

## Acknowledgements

We acknowledge the TB patients and their providers who provided the samples for this study and without which it would not have been possible. We acknowledge the ReseqTB team (Drs. Marco Schito and Matthew Esmundo) for providing us with data that allowed the validation of our GWAS hits. This study was supported by a biomedical research grant from the American Lung Association (PI MF, RG-270912-N), a K01 award from the BD2K initiative (PI MF, ES026835), and an NIAID U19 CETR grant (P.I. M.M., AI109755), the Belgian Science Policy (Belspo) (L.R., C.J.M.).

## Author contributions

M.F. and M.M. conceived this study, M.F. conducted the analysis and wrote the first version of the paper with key input from all authors. L.F. provided analysis support, and S.S. and M.M. provided analysis oversight. B.d.J., L.R. and C.J.M. curated, phenotyped, and sequenced the TDR isolates. A.S. and D.K. curated and tested the isolates from MSLI. D.v.S. curated and phenotyped the isolates from RIVM. J.S. and M.S. sequenced the isolates from MSLI and RIVM. R.C. cultured and maintained the archive of isolates from SES. J.S. and T.I. performed the sequencing and quality control on all SES isolates.

## Additional information

**Competing interests:** The authors declare no competing interests.

