## [Peer Review File · Nature Communications]

Reviewer #1 (Remarks to the Author):

The present manuscript presents a step forward the identification of new drug resistance determinants in Mtb. A key difference with previous publications is the use of detailed MIC data beyond the binary breakpoint resistant/susceptible. Genome-wide information on polymorphisms (coding, non-coding caused by SNP and indel) can then be statistically linked to MIC levels through GWAS analyses. The authors approach is novel as they use quantitative data and identifies, by using a validation dataset, a set of new loci involved in drug resistance. I think the value of the manuscript is not in the new resistant loci (we had some hints for several of them before) but on the idea that 1. detailed phenotypic data will likely be the way in the future to identify DR variants with weaker effect and 2. The architecture of resistance is much more complex than usually envisaged even for first line drugs like EMB. Overall is a very nice paper but I think more information can potentially be extracted or used to confirm the results. I have some comments I would like the authors to address:

1. The selection of strains is likely non-random. Even when the authors try to mitigate the effect by using a validation set most of the “discovery” strains (coming from Peru mainly) are likely “successful” strains, those particularly fit to be selected for sequencing. Is that is the case, the mutations identified are likely biased in the sense that the authors will tend to identify those more successful mutations (highly fit) rather than those more rare? At least because of the way they have been selected most mutations are likely to confer high resistance. Second bias is that there is enrichment on strains with known drug resistance mutations
2. Thus the overall number of rare mutations potentially identified and linked to relevant MIC is underestimated in the paper? I would like the authors to comment on this, do they think that an unbiased selection would have lead to a different SNP repertoire?
3. There are two aspects that the authors does not really address in my opinion, although I understand it is difficult. One is epistasis on MIC levels. How the presence of one mutation in one drug leads to different MIC effect of other mutation in another drug? I mean if you have one mutation for EMB with a given MIC X and another strain with a mutation in KAN with a given MIC Y. For those strains with both mutations at the same time, does X or Y changes? Is this one of the multiple reasons why there is missing explained variation? If yes, can the authors identify cases and analyze?
4. Is cross-resistance happening? Acquisition of resistance to one drug increasing the levels to other without additional changes? I understand, like the question above, this is really difficult to test as these are clinical strains with different genetic backgrounds but can the phylogeny be of use here by identifying strains with altered MIC and its closest relative but with no causative mutation for the particular drug ?
5. I though that high level resistance to EMB are usually a combination of ubiA + EMBB mutations, is that the reason why EMBAC does not have a strong phenotype? Can the authors test for co-existence of mutations and MIC levels (similar to point 2 but in this cases intra-drug rather than inter-drug MIC epistasis)?
6. Can the authors corroborate some of the results by evaluating homoplastic levels, while absence is not a proof, the presence of homoplasies use to be a good indicator as the authors have published before. I understand that we are talking about mutations with very low frequencies in the dataset but homoplasia maybe pooled by gene similar to the GWAS analysis?

7. I am surprised that the authors are not taken into account heteroresistance positions, if looking for rare causative mutations I think heteroresistance is a very good marker. In fact the % of times you see heteroresistance for your candidate genes and variants is a good indicator of on-going within host selection. Can the authors address this analysis, I think will reinforce the results. In general they should analyze heterozygous positions and if not used in the analysis the authors should reasoned why. They maybe behind some genotype-phenotype mismatches?
8. Are (some) of your intergenic variants hitting known or predicted regulatory regions, it can easily be checked with prediction programs or taking data from published RNAseq analysis? Is any involved in ncRNA regions? This will reinforce their functional role
9. Double promoter-coding mutations in *inhA* and *embB* – are you sure that the mutations are not phylogenetic, do they have an effect on MIC even if taken separately or it is just a specific genotype that by chance had one of the mutations not related to MIC and then acquired a second one related to MIC? As the strain selection is limited it may happen some phylogenetic clone is enriched in drug resistance giving a false positive? I understand you control for population structure but still it is very easy to show in a phylogeny whether all cases involving double mutations in the target are indeed mutations likely acquired independently and not part of uncontrolled phylogenetic background.

MINOR COMMENTS:

1. “Furthermore, the noncoding portion of the genome (10.3% by length) harbored a slightly disproportionate degree of variation with 13% of SNVs with an AF>0.05 occurring in these regions” I am not sure about how relevant is this or what the authors wants to transmit? While a portion of the noncoding genome will be under selection is not expected to harbour more diversity as the % under no selection is higher?
2. PPE35 -> how sure are you about the called variants in this gene?
3. 20% loci associated to resistance were intergenic - > this is a very interesting result and highlights that more attention should be put on the noncoding genome. I am wondering if mutations in intergenic are more likely associated to essential genes as oppose to non-essential genes that allow more coding mutations?
4. Discussion: “their association with higher levels of antibiotic resistance” -> higher than what? Please specify
5. Discussion: “inactivating protein mutations in drug targets” – Drug targets like *rpoB* cannot be inactivated, you mean “modifying” or something similar?
6. Please specify the complete SNP calling thresholds (coverage, qualities...etC)

Reviewer #2 (Remarks to the Author):

The study aims to provide a better understand drug resistance development in *M. tuberculosis*, which is global health threat. The authors combined genome sequencing data analysis with minimal inhibitory concentration testing to find new regions involved in resistance. The findings were then replicated in a second independent cohort.

While the results are potentially interesting, I have some methodological concerns that I raise in detail in my review.

In general, the paper is well written and the results are mainly clearly presented.

Results

The sample collection is not really clear to me. Please describe basic strain data e.g. phenotypic resistance in a supplemental table.

The MIC testing method and the selection for which drugs a strain was tested is also not clear to me. Is this a validated method? Has this been validated against a gold standard e.g. MGIT? What are the quality control measures? Did the authors use a susceptible and resistant strains for all drugs to check if the method is accurate?

As the genetic background of the isolates as well as dominant strain types may play an important role for the analysis performed, the author should do a bit more sophisticated analysis of this. So, please perform a high resolution phylogenetic strain classification and a cluster analysis. In addition, clustering should be taking into account when regions involved in drug resistance are determined.

Genomic analysis.

Accuracy of SNP calling is key for the analysis. How did the authors validate that their SNP finding workflow is accurate? Especially, for the variants in non-coding regions, a validation of the results is desirable.

New loci

The authors should be more carefully check the literature before stating the finding is novel. I just checked few and found the following:

WhiB6 was described before - or? E.g. Zeng et al. BMC genomics 2018

The association of ubiA and EMB resistance was already described by He et al. Tuberculosis 2015

ccsA was described by Ze-Jia Cui et al. Int J Mol Sci. 2016

This is not a convincing analysis

In general, the associations found need to be controlled for possible confounders e.g. dominant strain type by chance having that variant. In addition, presence of the variant in the intergenic regions needs also be confirmed.

It would also be great to see if the “new” variants occur alone in a single isolates causing resistance or if the occur together with another resistance marker.

Furthermore, it should be tested if those variants occur also in pan susceptible strains.

Proportion of variance in the resistance phenotype explained. This appears to be low for known resistance markers. Is this due to bad phenotypic data?

Replication cohort.

The results from the replication cohort are not really convincing. The cohort chosen needs to be described more in detail – the selection criteria need also to be present.

At the end, the authors were able to just confirm few variants from their analysis, several of them have already been described in the literature.

Discussion

The discussion needs to be revised according to the points made before. There actually not a lot that is discovered new. And the relevance for resistance development is not really clear. This should be put in a sensitive discussion. The MIC data are not really relevant for the major analyses performed. So, I am not so sure that these data really show. This represent an interesting part of the work, but needs to be better uses.

The variants in the intergenic regions need to be more thoroughly investigate and, as they are only few, may be confirmed by a secondary method e.g. PCR and sequencing.

Reviewer #3 (Remarks to the Author):

This study performed a GWAS on 1,452 clinical MTB isolates to evaluate genome wide associations between mutations in MTB genomic loci and drug resistance. The major findings of this paper are the novel associations between 13 genomic loci and drug resistance, validated using an independent dataset, and estimates for the heritability of resistance phenotypes to 11 anti-TB drugs.

The initial GWAS identified 50 loci associated with increased resistance to one or more antibiotics. In lieu of functional validation of these variants, the authors used an independent dataset to attempt to validate the mutations identified in the initial GWAS and confirmed 13 of these located in genes along with to intergenic regions.

A major difficulty with bacterial GWAS generally is the ability to accurately account for population structure and I am satisfied, based on the provided QQ plots, that the authors have done this. The details in the methods section, particularly those to do with GEMMA, should be expanded so as to make it possible to easily reproduce these results (reproducibility, to date, is another major problem in GWAS generally). It would be helpful to include a file with the commands used for all the analyses, not just the GWAS, in a supplementary file or else make the code available online through GitHub.

Apart from this, the only other changes I'd suggest are:

General changes:

Ensure consistency throughout for how numbers are displayed (1,526 versus 1526)

Use 'and' instead of &

Check the use of 'where' and 'were'

Use words instead of numbers for numbers less than ten

The paper could do with a general proof read as there are a few minor typos and grammatical errors included.

Specific changes:

Abstract

Add MIC and WGS in brackets after minimum inhibitory concentration and whole genome sequencing respectively

Introduction

Remove 'grim' from the sentence "...resulting in the grim reality of...". This seems too conversational to me

Please provide reference for the statement ending "...MTB critical concentrations are largely based on consensus and lack solid scientific support."

Give examples of second line drugs in sentence ending "...ethambutol and second line drugs"

Results

The use of the abbreviation SNV after single nucleotide substitutions; the 'V' stands for variant not substitution so change substitution to variant

Provide a meaning for AF (allele frequency?)

I suggest providing a slightly more detailed explanation of what a 'good' QQ plot should look like for those readers unfamiliar with them (this author has seen much worse than yours!)

Discussion

Provide a reference for the statement ending "...were previously associated with resistance in a prior GWAS albeit to non-AG agents"

I believe that the results are novel and of potential interest to the wider field and, as far as I'm aware, is the first study to use GWAS to identify genes and regulatory regions associated with particular MIC values in MTB.

Point by point response to reviewers' comments*:

*note we added R1, R2, R3 numbering to reflect reviewer 1, 2 and 3, and our responses are respectively labeled for ease of reference

Reviewer #1 (Remarks to the Author):

The present manuscript presents a step forward the identification of new drug resistance determinants in Mtb. A key difference with previous publications is the use of detailed MIC data beyond the binary breakpoint resistant/susceptible. Genome-wide information on polymorphisms (coding, non-coding caused by SNP and indel) can then be statistically linked to MIC levels through GWAS analyses. The authors approach is novel as they use quantitative data and identifies, by using a validation dataset, a set of new loci involved in drug resistance. I think the value of the manuscript is not in the new resistant loci (we had some hints for several of them before) but on the idea that 1. detailed phenotypic data will likely be the way in the future to identify DR variants with weaker effect and 2. The architecture of resistance is much more complex than usually envisaged even for first line drugs like EMB. Overall is a very nice paper but I think more information can potentially be extracted or used to confirm the results. I have some comments I would like the authors to address:

R1.1 The selection of strains is likely non-random. Even when the authors try to mitigate the effect by using a validation set most of the “discovery” strains (coming from Peru mainly) are likely “successful” strains, those particularly fit to be selected for sequencing. Is that is the case, the mutations identified are likely biased in the sense that the authors will tend to identify those more successful mutations (highly fit) rather than those more rare? At least because of the way they have been selected most mutations are likely to confer high resistance. Second bias is that there is enrichment on strains with known drug resistance mutations. Thus the overall number of rare mutations potentially identified and linked to relevant MIC is underestimated in the paper? I would like the authors to comment on this, do they think that an unbiased selection would have lead to a different SNP repertoire?

A1.1 The reviewer raises an important point. We are not sure if identifying mutations that are more fit results in bias *per se* or is in fact a feature of studying clinical pathogen samples. We propose that the relevant variants for clinical and diagnostic purposes are those that are more fit and hence likely to propagate between clinical hosts and exist at relevant frequencies. The isolates studied were sampled in three different batches: 1- sampled to over-represent drug resistance from an older archive of isolates from Peru collected 1997-2004, 2- sampled from a longitudinal household contact study of TB in Northern lima conducted between 2009 and 2012 and 3- sampled from labs in the Netherlands, Belgium and the Massachusetts State lab archives because of the availability of MIC data (and as a result also over represent resistant isolates). We agree that there is an overrepresentation of samples and mutations within that cause higher levels of resistance as a result, this was by design to increase the power of the analysis and also a function of our interest in clinically relevant resistance. We do not expect however an enrichment of strains with known resistance mechanisms as there was no selection based on the genotype *per se*. We did observe a high degree of genotypic complexity despite the selection of more resistant isolates, but we agree that it is possible that this complexity may still be underestimated. We agree that future work can focus and enrich samples with clinically susceptible

isolates to examine variants within those populations that affect MIC (and perhaps explain the 'wild type' MIC distribution). However these analyses will likely require many thousands of isolates as the effects are expected to be much smaller than what we observed in this sample. Conducting agar based MIC testing at those scales for MTB is currently prohibitively expensive and time consuming, and MIC's are not done for routine clinical care. We have now added the following text in the discussion to elaborate on the potential consequences of the sampling procedure we undertook.

“To achieve sufficient statistical power to detect associations between genetic variation and clinically relevant resistance phenotypes we oversampled isolates that had higher levels of drug resistance. It is possible that this over-representation of high level resistance enriched for a subset of genetic variants and make it less likely to capture variants that are rarer or have smaller effects.”

R1.2 There are two aspects that the authors does not really address in my opinion, although I understand it is difficult. One is epistasis on MIC levels. How the presence of one mutation in one drug leads to different MIC effect of other mutation in another drug? I mean if you have one mutation for EMB with a given MIC X and another strain with a mutation in KAN with a given MIC Y. For those strains with both mutations at the same time, does X or Y changes? Is this one of the multiple reasons why there is missing explained variation? If yes, can the authors identify cases and analyze?

A1.2 With regards to mutation-mutation interactions and their effect on individual drug MICs, we examined this only partially by assessing the interaction between the MTB lineage background (lineage 2 vs lineage 4) and 5 common resistance genetic variants namely: *katG* S315T, *rpoB* S450L, *embB* M306V, *inhA* -15, *pncA* H51R. We found no difference in MIC distributions between lineage 2 and lineage 4 isolates harboring each of these variants. This may have been related to the large effect that each of these mutations had. In response to reviewer 1's comments (R1.2 and R1.4) we have now extended this analysis to include an assessment of an interaction between *rpoB* D435V variant known to have a smaller effect on rifamycin MIC (See Sirgel et al 2012)¹ we found there to be a difference in rifabutin MIC distribution between lineage 2 and lineage 4 isolates harboring this mutation. As the reviewer suggests we now also examined interactions between *embB* M306V (causative of resistance to EMB) and *rrs* A1401G (causative of resistance to Kanamycin), as well as between *embB* and *ubiA*, *embA* and *embC* (see below A1.4). We have added the description of this analysis to the result section that currently reads as follows:

“We sought to determine if there are detectable interactions between specific resistance sites, genes and the genetic lineage. We hypothesized that because antibiotic resistance arises as a result of strong positive selection in MTB and several sites have large effects on the phenotype that the MIC distributions observed for such resistance mutations would not vary appreciably across different lineages. We focused on lineage 4 and lineage 2, as they were well represented in our sample. Examining the six mutations: *katG* S315T, *rpoB* S450L, *rpoB* D435V, *embB* M306V, *inhA* -15, *pncA* H51R for INH, RIF, RFB, EMB, ETA and PZA respectively, we found the MIC distributions to not to be appreciably different for five of the six mutation-drug pairs (P-value >0.2). We did associate the mutation *rpoB* D435V with higher median rifabutin MICs

among lineage 2 isolates than among lineage 4 (median 0.375mg/L vs. ≤ 0.125 P-value 6×10^{-4}). Examining interactions between specific pairs of mutations and genes, we first tested if the acquisition of additional resistance mutations causative of resistance to other drugs is associated with increases in MIC. We focused on the first line drug EMB for which the most significant novel hit *ubiA* was identified as well as the second line aminoglycoside KAN. We examined the loci *embB*, *embA*, *embC* and *ubiA* for EMB and *rrs* for KAN. We found EMB MIC levels to be higher among isolates with both an A1401G *rrs* variant and an M306V *embB* variant, as compared with those with M306V *embB* and without the *rrs* variant (P-value 0.005 median >15 (IQR 12.5 to 15) vs >15 (IQR >15 to >15). *UbiA* and *embA* variants were also more common among isolates with both M306V *embB* and A1401G *rrs* compared with isolates harboring only the former (*embA* OR 13.8 (95% CI 6.1-33.6, P-value $<10^{-12}$), *ubiA* OR 6.7 (95% CI 3.3-13.8, P-value $<10^{-8}$). After excluding isolates with *embA* and *ubiA* variants, isolates with both the *rrs* and *embB* variant still tended to have a higher MIC but the P-value decreased to 0.05. Isolates with both *embA* and *embB* variants were more likely to have a higher EMB MIC (median >15 , IQR 7.5 to >15) than those with only *embB* variants (median >15 , IQR 3.5 to >15 , P-value 4×10^{-4}). The co-occurrence of *embB* and *ubiA* variants was also associated with elevations in the EMB MIC relative to *embB* variants alone: median of >15 , IQR 7.5 to >15 to a median of >15 , IQR 12.5 to >15 , P-value 6×10^{-4} . On the other hand, variants in *embC* were not more likely to co-occur with *embB* variants, P-value 0.6; and there was no difference in EMB MIC between isolates with both *embC* and *embB* variants vs. those with only *embB* variants, P-value 0.5.”

And expanded the section in the discussion that now reads as follows:

“We did examine a set of specific interactions between six canonical resistance mutations and genetic lineage (lineage 4 vs 2), and between variants in the loci *embABC* and *ubiA* one of the novel candidates. Although for several of the loci we examined that individually had large effects, like *katG* S315T and *rpoB* S450L, we could not detect an interaction with the genetic background, in the case of *ubiA*, *embA* and *rpoB* D435V we found evidence for the presence of at least additive interactions on the drug MIC. It thus seems likely that such interactions exist for other mutations, and may be widespread in bacterial genomes, especially between variants with smaller effects. These conclusions are consistent with prior evidence from allelic exchange experiments³⁶.”

1. Sirgel, F. A. *et al.* *gyrA* mutations and phenotypic susceptibility levels to ofloxacin and moxifloxacin in clinical isolates of *Mycobacterium tuberculosis*. *J. Antimicrob. Chemother.* **67**, 1088–1093 (2012).

R1.3 Is cross-resistance happening? Acquisition of resistance to one drug increasing the levels to other without additional changes? I understand, like the question above, this is really difficult to test as these are clinical strains with different genetic backgrounds but can the phylogeny be of use here by identifying strains with altered MIC and its closest relative but with no causative mutation for the particular drug ?

A1.3 As the reviewer suggests there is cross-resistance between drugs of the same class or mechanism of action, for example between INH and ETH, RIF and RFB & between AMI, KAN and CAP. There is strong evidence supporting the occurrence of cross resistance in the prior

literature on this topic for MTB for these drugs. In the work under consideration here, we confirm these associations, as we make associations between several resistance loci and multiple drugs in our GWAS: namely between the *inhA* promoter with both INH and ETH, *rpoB* with both RIF and RFB, and *rrs* with AMI, KAN or CAP as detailed in Table 1 and Supplementary Tables 2 and 3. Although we observe these associations, it is challenging to confidently make claims about cross resistance *between* drug classes for example *embB* variants and INH vs EMB because of the co-linearity of resistance in TB. Co-linearity, which is at least in part related to how TB medications are used clinically, is discussed in the limitations paragraph in the discussion section. We have now expanded this section to discuss cross resistance, and currently reads as follows:

“Given the recognized step wise acquisition of resistance in MTB², it is very challenging to determine accurately which drug resistance is in fact associated with a particular gene or genetic region. For example resistance to any of the second line agents, fluoroquinolones like MXF, SLI’s like AMI, or to first line agents like PZA and EMB, nearly always co-exists with resistance to INH and RIF, and it is thus not possible to perform association conditioning on the absence of resistance to those agents. This also significantly limits our ability to assess for mutations that can result in cross-resistance between drug classes; however our results do support recognized associations between genetic loci and drug members of the same class such as *inhA* promoter variants with INH and ETH resistance, *rrs* variants and AMI, KAN or CAP resistance and *rpoB* variants and RIF or RFB resistance (Supplementary Tables 2 and 3). “

R1.4 I thought that high level resistance to EMB are usually a combination of *ubiA* + EMBB mutations, is that the reason why EMBAC does not have a strong phenotype? Can the authors test for co-existence of mutations and MIC levels (similar to point 2 but in this cases intra-drug rather than inter-drug MIC epistasis)?

A1.4 Our inability to make an association between *embA* and ethambutol (EMB) resistance is at least in part related to the fact that *embA* variants not previously determined to be lineage markers (lineage marker as assessed in Walker *et al* Lancet ID 2015) were less common in our test set than *ubiA* variants. Specifically we found only 3.4% of isolates to harbor an *embA* variant in contrast with 7.1% for *ubiA*. In addition the point estimate for marginal effect of *embA* variants on EMB resistance was 4 times as low as that measured for *ubiA*; the former with logMIC change of 0.129 and not significant (P-value of 0.24). For *embC*, variants were more common, occurring in 9.9%, but we nevertheless found no association between *embC* and EMB MIC (logMIC change -0.084 standard error= 0.128, P-value of 0.514). We assessed if nevertheless an effect of *embA* and *embC* was mediated via interaction with *embB* variants rather than directly. We found that whereas *embA* variants were more common among isolates with *embB* variants (OR 2.47 Fisher 95% CI 1.52-4.11), *embC* variants were not (OR 1.06 Fisher 95% CI 0.82-1.39). In addition isolates with both *embA* and *embB* variants were more likely to have a higher EMB MIC (median >15, IQR 7.5 to >15) that those with only *embB* variants (median >15, IQR 3.5 to >15); Wilcoxon rank sum test P-value of 4×10^{-4} . Variants in *ubiA* were even more likely to co-occur with *embB* variants (OR 7.02 Fisher 95% CI 4.52-11.31), and elevate the EMB MIC relative to isolates with *embB* variants alone from a median of >15, IQR 7.5 to >15 to a median of >15, IQR 12.5 to >15, Wilcoxon P-value 6×10^{-4} . There was no difference in EMB MIC between isolates with both *embC* and *embB* variants vs. those with only *embB* variants

Wilcoxon P-value of 0.5. We now describe these additional analyses in the manuscript's results section as outlined in **A1.2** above.

R1.5 Can the authors corroborate some of the results by evaluating homoplastic levels, while absence is not a proof, the presence of homoplasies use to be a good indicator as the authors have published before. I understand that we are talking about mutations with very low frequencies in the dataset but homoplasmy maybe pooled by gene similar to the GWAS analysis?

A1.5 We provide a detailed breakdown of homoplasmy, distribution of variants by lineage, within the candidate loci in supplementary table 5. In addition and based on the reviewer recommendation, we have now also run a locus-level homoplasmy detection software tool called TreeWAS (Collins et al. PLoS Computational biology 2016). The results of this are now described in Supplementary Table 6. Of our validated loci, we found *whiB6*, *PPE35*, *Rv2752c*, *Rv3236c*, *ubiA*, *thyX-hsdS.1* to show significant homoplasmy at a permutation P-value <0.005 . Notably these were the same loci that displayed more diverse variation across the length of locus see Supplementary Figure 3. We have now added this to the results section which now reads:

“In a formal test for phylogenetic convergence²², *ubiA*, *whiB6*, *Rv2752c*, *PPE35*, *Rv3236c*, and *thyX-hsdS.1* displayed significant homoplasmy at a permutation P-value <0.005 (Table S6).“

We have also expanded the methods section to describe this:

“In a parallel analysis we ran a test for phylogenetic convergence¹¹ using the MEGA5 NJ tree (Figure S4) and the treeWAS R package²² utilizing the simultaneous score and a permutation P-value threshold of <0.005 to assess significance.“

R1.6 I am surprised that the authors are not taken into account heteroresistance positions, if looking for rare causative mutations I think heteroresistance is a very good marker. In fact the % of times you see heteroresistance for your candidate genes and variants is a good indicator of ongoing within host selection. Can the authors address this analysis, I think will reinforce the results. In general they should analyze heterozygous positions and if not used in the analysis the authors should reasoned why. They maybe behind some genotype-phenotype mismatches?

A1.6 We agree with the reviewer that heteroresistance, *i.e.* heterozygous calls for an allele within one sample, can explain some phenotype-genotype discrepancies. Because of this we included all variants that otherwise met quality criteria that had a purity of $>40\%$, *i.e.* 40% or more of the reads support the presence of this call. This is in contrast to the usual threshold of higher purity often $>75\%$ or $>90\%$ used in many studies or is the default of several variant callers. We chose the 40% cut-off to avoid heterogeneity that results from sequencing error and PCR bias. In a study we previously published, that focused on 28 MTB resistance loci we found that lowering the threshold from 40% to 10% did not appreciably improve the sensitivity of resistance prediction (*i.e.* improve concordance between genotype-phenotype), but decreased the specificity of prediction (Farhat et al. AJRCCM 2016). We have now added justification for this threshold in the methods section which now reads as follows:

“We used the purity threshold of 0.4 as in a previous comparison with a lower threshold (of 0.1) there was no significant improvement in sensitivity over specificity¹⁸.”

R1.7 Are (some) of your intergenic variants hitting known or predicted regulatory regions, it can easily be checked with prediction programs or taking data from published RNAseq analysis? Is any involved in ncRNA regions? This will reinforce their functional role

A1.7 Mutations in the intergenic region upstream of *thyX* have been shown to modulate *thyX* expression in the literature cited in the discussion (Zhang et al 2013). We observed the same mutation occurring at coordinate -9 from the start of the *thyX* gene, genomic coordinate 3,067,954 A>T in our study, in addition to several adjacent variants at position -16 and -4 that occurred in 37 and 4 isolates respectively. For the intergenic region upstream of *espK*, the transcription start site (TSS) was mapped by Shell et al 2015 to position -224 from the beginning of the *espK* gene (genomic coordinate 4,360,006). We observed 18 isolates to harbor five different variants within 55bp from the TSS with the closest variant occurring 9bp from the TSS. We have now added this assessment to our results section which currently reads as below. We also now highlight the predicted TSS in Figure S3. We were not able to find evidence that these variants alter any ncRNA, but to our knowledge data on ncRNA regions for MTB is currently limited.

“For the intergenic hits, we observed a concentration of variants around the predicted transcriptional start site in both cases (Figure S3)”

R1.8 Double promoter-coding mutations in *inhA* and *embB* – are you sure that the mutations are not phylogenetic, do they have an effect on MIC even if taken separately or it is just a specific genotype that by chance had one of the mutations not related to MIC and then acquired a second one related to MIC? As the strain selection is limited it may happen some phylogenetic clone is enriched in drug resistance giving a false positive? I understand you control for population structure but still it is very easy to show in a phylogeny whether all cases involving double mutations in the target are indeed mutations likely acquired independently and not part of uncontrolled phylogenetic background.

A1.8 For this analysis, we specifically looked at genetic variants previously associated with resistance, and compared the effect of variants at the most common promoter site with the effect of the variants at the most common codon site in the listed loci on the drug MIC. Because of the choice of variants, none were phylogenetically restricted. Below is a table providing the lineage breakdown of these variants, and given the number of different sub-lineages affected, this confirms that none were lineage related/related to a specific phylogenetic clone. We thus conclude that either by chance or through a process of adaptation to continued drug pressure multiple mutations may accumulate simultaneously in the promoter region and the gene body. We have now also included the below table in the manuscript as Supplementary Table 6.

We have now also altered the last paragraph in the results section to read:

“We focused on the codon and promoter site with the largest allele frequency in each case. Isolates not infrequently had both a gene body and a promoter mutation: 12% of isolates with *embB* promoter mutations also had an *embB* codon 306V, and 18% of isolates with an *inhA*

promoter mutation also had a mutation at *inhA* codon 21. None of these variants were phylogenetically restricted (Table S6).”

Lineage	Total count	embAB promoter variants (any of the below)			embB coding variant
		C16T promoter embB	C16G promoter embB	C16A promoter embB	M306V embB
1.1.1	3	0	0	0	0
1.1.2	5	0	0	0	0
1.1.3	2	0	0	0	0
1.2.1	5	0	0	0	1
1.2.2	4	0	0	0	1
2	9	0	0	0	0
2.1	1	0	0	0	0
2.2	24	0	0	0	8
2.2.	140	1	2	1	41
3	23	0	0	0	2
4	171	5	4	0	12
4.1	447	4	2	3	26
4.2.1	6	0	0	0	2
4.2.2	6	1	0	0	1
4.3	584	2	9	5	87
4.5	2	0	0	0	0
4.6.1	21	0	0	0	13
4.7	11	0	0	0	0
4.8	20	0	0	0	1
4.9	1	0	0	0	0
5	1	0	0	0	0

Lineage	Total count	inhA	inhA codon 21 variants		
		promoter variant C15T promoter inhA	I21M inhA	I21V inhA	I21T inhA
1.1.1	3	0	0	0	0
1.1.2	5	0	0	0	0
1.1.3	2	0	0	0	0
1.2.1	5	3	0	0	0
1.2.2	4	0	0	0	0
2	9	0	0	0	0
2.1	1	0	0	0	0
2.2	24	3	0	0	0
2.2.	140	16	0	2	0
3	23	3	0	0	0
4	171	8	0	0	0
4.1	447	62	0	1	3
4.2.1	6	3	0	0	0
4.2.2	6	0	0	0	0
4.3	584	116	1	1	35
4.5	2	0	0	0	0
4.6.1	21	0	0	0	0
4.7	11	2	0	0	0
4.8	20	4	0	0	0
4.9	1	0	0	0	0
5	1	0	0	0	0

MINOR COMMENTS:

R1.9 “Furthermore, the noncoding portion of the genome (10.3% by length) harbored a slightly disproportionate degree of variation with 13% of SNVs with an AF>0.05 occurring in these regions” I am not sure about how relevant is this or what the authors wants to transmit? While a portion of the noncoding genome will be under selection is not expected to harbour more diversity as the % under no selection is higher?

A1.9 In response to the reviewer comment we have now removed this sentence.

R1.10 PPE35 -> how sure are you about the called variants in this gene?

A 1.10 Variants in PPE35 were confirmed by visualization using IGV v2.4.9 and were manually inspected to confirm their accuracy. We also simulated Illumina reads originating from a reference genome containing variants in PPE35 and remapped these to the same reference genome or a different MTB reference genome and showed that the false positive rate for variants at the same positions we detected in our study was <1%. We are preparing a description of these results to be submitted soon for peer review in another manuscript.

R1.11 20% loci associated to resistance were intergenic -> this is a very interesting result and highlights that more attention should be put on the noncoding genome. I am wondering if mutations in intergenic are more likely associated to essential genes as oppose to non-essential genes that allow more coding mutations?

A1.11 We explored the possibility that resistance variants occurring in promoter regions were more likely to be associated with essential genes. The following 11 loci are well recognized resistance determinants with variants occurring within the gene body: *katG*, *rpoB*, *rrs*, *rpsL*, *gyrAB*, *gid*, *rpsA*, *thyA*, *tlyA*, *embC*, and *ethA*. All of these except *embC* were associated with resistance in our study. Of the 11, 7 are essential genes, and 4 are non-essential namely *gid*, *tlyA*, *katG*, and *ethA* according to DeJesus et al 2017 (accessed through Mycobrowser). Of the 7 resistance variants where variants can occur or occur exclusively in the promoter region in our study namely: *eis*, *pncA*, *inhA*, *ahpC*, *thyX*, *espK*, and *embAB*, 4 are non-essential namely *eis*, *pncA*, *ahpC* and *espK*. Thus there is a slightly higher but non-significant proportion of non-essentiality among loci where promoter variants are associated with resistance. We conclude that the available essentiality measure is not a good measure of the functional impact of variants within a gene. We have now added a sentence on this to the last paragraph of the results sections that reads as follows:

“We also tested the possibility that genes that harbor promoter variants associated with resistance were more likely to be essential genes than genes that exclusively harbor variants in the gene body in association with resistance. Of the latter, 7/11 were essential whereas only 3/7 genes with promoter resistance variants were essential suggesting that the gene essentiality measure is limited in assessing the functional impact of a variant.”

R1.12 Discussion: “their association with higher levels of antibiotic resistance” -> higher than what? Please specify

A1.12 We deleted ‘higher levels of’, sentence now reads ‘association with antibiotic resistance’

R1.13 Discussion: “inactivating protein mutations in drug targets” – Drug targets like rpoB cannot be inactivated, you mean “modifying” or something similar?

A1.13 Changed to ‘protein modifying’

R1.14 Please specify the complete SNP calling thresholds (coverage, qualities...etc)

A1.14 We now provide all SNP calling thresholds in the methods section under variant calling. The text reads as follows:

“Genome coverage was assessed using SAMtools 0.1.18⁵³ and FastQC⁵⁴ and read mapping taxonomy was assessed using Kraken⁵⁵. We aligned the Illumina reads to the reference MTB isolate H37Rv NC_000962.3 using Stampy 1.0.23⁵⁶ and variants were called by Platypus 0.5.2⁵⁷ using default parameters. Strains that failed sequencing at a coverage of less than 95% at $\geq 10x$ of the known drug resistance regions, or that had a mapping percentage of less than 90% to *M. tuberculosis* complex were excluded. Genomic regions not covered at $\geq 10x$ in at least 95% of the remaining isolates were filtered out from the analysis, i.e. no attempt at association with variants in those regions was made. In the remaining regions, variants were further filtered if they had a quality of < 15 , purity of < 0.4 or did not meet the PASS filter designation by Platypus. We used the purity threshold of 0.4 as in a previous comparison with a lower threshold (of 0.1) there was no significant improvement in sensitivity over specificity¹⁸. We also excluded any indels $> 3bp$ in size or large sequence polymorphisms. Further quality control was performed after genome wide association when associated PE/PPE gene and indels were visualized and manually inspected using IGV v2.4.9⁵⁸.”

Reviewer #2 (Remarks to the Author):

The study aims to provide a better understand drug resistance development in *M. tuberculosis*, which is global health threat. The authors combined genome sequencing data analysis with minimal inhibitory concentration testing to find new regions involved in resistance. The findings were then replicated in a second independent cohort.

While the results are potentially interesting, I have some methodological concerns that I raise in detail in my review.

In general, the paper is well written and the results are mainly clearly presented.

Results

R2.1 The sample collection is not really clear to me. Please describe basic strain data e.g. phenotypic resistance in a supplemental table.

A2.1 We now highlight all details of sample collection in the first section of the online methods titled “Sample Collection” which reads as below. In addition the full line by line MIC data for drugs is given in Table S1.

“MTB sputum based culture isolates were selected from (1) a Peruvian patient archive of culture isolates enriched for resistance based on prior targeted resistance gene sequencing and binary DST phenotype¹⁸ (n=496), or (2) sampled from a longitudinal cohort of patients with Tuberculosis from Lima Peru⁴⁷ enriched for multidrug resistance based on prior binary DST (n=568). These 1,064 isolates had phenotypic resistance testing by MIC for 12 drugs repeated (see below) at the National Jewish Hospital (NJH) Denver, CO, and underwent whole genome sequencing. Data from these isolates were pooled with data from two additional samples: a convenience sample from three national or supranational reference laboratories selected based on the availability of MIC data: the Institute for Tropical Medicine -Antwerp, Belgium, the Massachusetts State TB Reference Laboratory -Boston, MA, and the National Institute for Public Health and the Environment-Bilthoven, Netherlands (n=411) and a sample of 83 pan-susceptible isolates from the Peruvian TB cohort⁴⁷ added to increase the representation of sensitive isolates.”

R2.2 The MIC testing method and the selection for which drugs a strain was tested is also not clear to me. Is this a validated method? Has this been validated against a gold standard e.g. MGIT? What are the quality control measures? Did the authors use a susceptible and resistant strains for all drugs to check if the method is accurate?

A2.2 According to the Clinical Laboratory Standards Institute (CLSI) reference on susceptibility testing of Mycobacteria, the gold standard for MIC measurement is the indirect proportions method on Middlebrook agar for all drugs except pyrazinamide. For pyrazinamide the gold standard approach is to use a radiometric approach/ the MGIT platform. We used these gold standard methods on 7H10 agar and MGIT 960 respectively. We now highlight this in the online methods section and Supplementary Tables 8 and 9. Critical concentrations were according to CLSI and WHO recommendations, in addition we tested more finely spaced concentrations between the critical concentration and achievable serum concentration according to Alsultan and Peloquin 2014 (citation 50 in the text). The testing at the National Jewish Laboratory had rigorous quality control measures in place including testing of susceptible and resistance reference strains. Please refer to our response to Reviewer comment R2.8 for more details on this.

R2.3 As the genetic background of the isolates as well as dominant strain types may play an important role for the analysis performed, the author should do a bit more sophisticated analysis of this. So, please perform a high resolution phylogenetic strain classification and a cluster analysis. In addition, clustering should be taking into account when regions involved in drug resistance are determined.

A2.3 We measure strain-strain relationships with a co-variance/genetic relatedness matrix that measures strain similarity in a continuous manner. The control for population structure/dominant strain types using such a genetic relatedness matrix has gained popularity very recently, see Coll et al Nature Genetics 2018 for TB and Li Y et al. Nature Communications 2019 for streptococcus. We use the co-variance matrix in a linear mixed model to perform the genome wide association as was done in both of these publications. This allowed us to correct for population structure/phylogenetic relationships for each region we associate. It down weights signal from regions that are phylogenetically restricted by inflating the variance of the coefficient of association. This approach has been used successfully in the two studies cited above and

others cited in the text, but we agree that it may still be possible that there is residual confounding from population structure. Because of this, we confirm the correction of population structure using QQ-plots (Supplementary Figure 2). As described by Reviewer 3 below our QQ show that population structure is largely eliminated using the GRM and LMM approach. We further validate our hits in an independent data set that has a different lineage distribution, to address the known problem of limited replication of GWAS findings previously in the literature. We also provide evidence of convergence and homoplasy of our final hits in Supplementary Tables 5 & 6.). In the latter we also report on the results of a phylogenetic convergence test, run in response to reviewer 1's comments that shows considerable overlap with our validated hits. We also now provide, in addition to the heatmap showing isolate diversity/classification (Figure 1c), a phylogenetic tree to show the diversity of our sample set (new Figure S4). Finally, we have now added additional detail to the methods section to clarify the approach to population structure and the running of a parallel phylogenetic convergence test.

“Genotype, GWAS and control for population structure: ... We controlled for population structure by computing a genetic relatedness matrix (GRM) that measures genetic similarity as the co-variance between the individual isolate genetic variant vectors. For the GRM computation we included all synonymous and non-synonymous SNPs and indels but excluding variants in known drug resistance loci and variants occurring at a MAF of <0.01 using the software package GEMMA⁶⁴. Genome wide association was performed using a linear mixed model...”

Genomic analysis.

R2.4 Accuracy of SNP calling is key for the analysis. How did the authors validate that their SNP finding workflow is accurate? Especially, for the variants in non-coding regions, a validation of the results is desirable.

A2.4 We used validated software for read classification, mapping and variant calling cited in the methods section. We've also previously used this data pipeline and validated against both targeted sequencing and Sanger sequencing (Farhat et al AJRCCM 2016). We also provide all our data and thresholds within the manuscript to the community of readers to facilitate replication. For non-coding regions and PPE genes, we further confirmed the calls by manual inspection and visualization using IGV.

New loci

R2.5 The authors should be more carefully check the literature before stating the finding is novel. I just checked few and found the following:

WhiB6 was described before - or? E.g. Zeng et al. BMC genomics 2018

The association of ubiA and EMB resistance was already described by He et al. Tuberculosis 2015 cccsA was described by Ze-Jia Cui et al. Int J Mol Sci. 2016. This is not a convincing analysis

A2.5 Unfortunately there is no public list of 'novel' loci that is maintained by the research community on the topic. We thank the reviewer for providing us with those three references. The Zeng et al paper describes an association between whiB6-Rv3863 and resistance but not whiB6 itself. We have now added the two references to our discussion section. (He et al is reference 28, ZJ Cui et al is reference 37) in the following two contexts:

“...mutations introduced at *ubiA* codon 237 were shown to increase gene function and elevate decaprenylmonophosphoryl-B-D-ribose or arabinose (DPA) levels^{27,28}”

“*CcsA* and *katG* variants commonly co-occurred in one genomic analysis of 288 isolates from China and the pairs were found to be significantly associated with resistance³⁷.”

We also added the Zeng et al reference to Supplementary Table 8 our meta-analysis table of TB GWAS hits.

We defined ‘novel’ for the purposes of our analysis as regions not currently tested for in commercial molecular tests for drug resistant TB. We apologize on the lack of clarity on this. We have now added a sentence describing this in the methods section under the title ‘Definition of known and novel drug resistance loci’. We attempted our best to identify loci that have been previously reported and performed a meta-analysis of the three prior main GWAS studies of MTB drug resistance (Supplementary Table 8).

“Definition of known and novel drug resistance loci

We define the MTB known resistance loci as the following genes *katG*, *inhA* & its promoter, *ahpC* promoter, *kasA*, *rpoB*, *embA*, *embB*, *embC* & *embA-embC* intergenic region, *ethA*, *gyrA*, *gyrB*, *rrs*, *rpsL*, *gid*, *pncA* & its promoter, *tlyA*, *thyA*, *rpsA*, *eis* promoter and the compensatory genes *rpoC*, *rpoA* based on prior published work^{6,16,20–25} and the use of many of these regions in commercial molecular diagnostics for MTB. We define loci other than those listed above as ‘novel’ loci if they were found to be significantly associated in the GWAS.”

R2.6 In general, the associations found need to be controlled for possible confounders e.g. dominant strain type by chance having that variant. In addition, presence of the variant in the intergenic regions needs also be confirmed.

A2.6 We agree with the reviewer than control for population structure and dominant strain type is paramount in genome wide association studies. Please refer to our response to R2.3 and R2.4 for the approach we have taken.

R2.7 It would also be great to see if the “new” variants occur alone in a single isolates causing resistance or if the occur together with another resistance marker. Furthermore, it should be tested if those variants occur also in pan susceptible strains.

A2.7 The variant breakdown by resistance is given in Supplementary Table 5. We have now expanded this to include the frequency among pan-susceptible isolates. Although the most common variant for some loci is observable among a low proportion of INH and RIF susceptible isolates (0-11%), it is notable that this breakdown ignores the MIC variability on both sides of the critical concentration, which the regression model captures. We believe strongly that the strength of the regression approach is in its ability to make associations between genetic variants and smaller changes in MIC that may not yet be clinically evident. We also examined the co-occurrence of the hit variants with known resistance makers in the detail in the case of *ubiA*,

embA, embB, and embC in our response to comment R1.4 above and now added to the manuscript. We also tackle this question at a genome scale by measuring differences in variance explained by the hit loci and the known resistance regions. We measure that, on average, the novel loci only explain an additional 1.5% of the resistance phenotypic variance, in contrast to 10% on average for the known regions. This emphasizes that genetic mechanism underlying MIC variability are complex, with many individually rare mutations contributing. We discuss this in the manuscript in the section quoted below, now modified from previously:

“We estimate that 64-88% of the MIC variance to be explained by genetic effects, with standard errors ranging from 2-6%. The remaining proportion may be explained by other factors such as genetic interactions, mutation heterogeneity or environmental or other testing related factors that result in MIC level variability. It is notable that we found the known resistance loci to explain a relatively low amount of the total variation ranging as low as 0.01 for ETA to 0.24 for AMI. The gap between total PVE and that attributable to known drug resistance loci, is not completely explained by the presence of the novel genetic loci as these explained an even lower proportion than known drug resistance loci, likely related to their low mutation frequency. This gap may be better explained by lineage or gene-gene interactions. We did examine a set of specific interactions between six canonical resistance mutations and genetic lineage (lineage 4 vs 2), and between variants in the loci embABC and ubiA one of the novel candidates...”

The rest of this text is quoted above under R1.2.

R2.8 Proportion of variance in the resistance phenotype explained. This appears to be low for known resistance markers. Is this due to bad phenotypic data?

A2.8 Several steps were taken to assure the quality of the phenotypic data. This included two types of experimental controls that were performed during the course of MIC measurement. The below two paragraphs are now added to the methods section under *Culture and Drug resistance/MIC testing* describing the quality control procedures which consisted of the following two main measures.

- 1- “The repeat testing, with every batch, of two ‘**external control**’ MTB strains, one resistant to all drugs except moxifloxacin and linezolid and the other susceptible to all drugs. Testing of these two reference strains was repeated using the same method (indirect proportions on 7H10 agar) with each batch of ~30 clinical isolates to confirm the reference MICs replicate at the exact level. If either or both of the reference strains failed to replicate the expected MIC for one or more drugs, the whole batch of isolates was retested. Of the 33 batches and replicate reference MTB strain tests conducted during the 2.5 years of testing, only 1 failed to replicate and was repeated. The reference MICs were reproduced upon the repeat testing.
- 2- ‘**Internal controls**’: Every clinical isolate was tested on an agar plate split into 4 quadrants, 3 quadrants contained increasing concentrations of the drug and the 4th was a control quadrant free of drug. If the isolate failed to grow in the control quadrant (at least 50 colonies), the isolate was re-tested (*i.e.* on a new plate) given the concern for inadequate inoculum. The culture plates were also monitored for contamination. If any contamination was observed the isolate was also retested. Over 29/1091 isolates were

retested because of there was either no-growth or contamination. Two isolates of the 29 could be grown and MICs for these isolates were run twice repeated. In both cases the MICs were replicated to within 1 MIC dilution.”

R2.9 Replication cohort.

The results from the replication cohort are not really convincing.

The cohort chosen needs to be described more in detail – the selection criteria need also to be presented. At the end, the authors were able to just confirm few variants from their analysis, several of them have already been described in the literature.

A2.9 We have now expanded Supplementary Table 4 that describes the validation dataset by adding the detailed resistance phenotype. All accession numbers are all provided there, and formatted data and code given to allow readers to replicate the analysis. The replication cohort was a convenience sample of available high quality public MTB genomic and drug resistance data. We did not select these isolates based on their resistance profile or their lineage. We now highlight this for added clarity in the methods section. The majority of the data was curated by the Reseq TB collaboration (cited in the manuscript). We also added data from two prior publications: Gardy et al NEJM 2011 and Zhang et al Nat Gen 2013. The validation data selection in the methods section now reads as follows:

“We validated the genomic regions identified above in an independent public dataset with binary phenotype data. The validation dataset consisted of a convenience sample of 792 MTB isolates obtained by pooling data from the ReSeqTB knowledge base (<https://platform.reseqtb.org/>)⁴² with additional MTB whole genome sequences and phenotype data curated manually from two additional references^{26,65} (Table S4). We did not select isolates for the validation set based on lineage or drug resistance profiles.“

There are several reasons why hits found in the test set could not be replicated in the validation set. At the forefront is the low allele frequency for the majority of the candidates. This coupled with the use of a validation set with half the test set sample size and with only 35% MDR isolates significantly reduced statistical power. We also did not have MICs available for these isolates and instead used binary resistance phenotypes. This further impacts power. Further we used a stringent cutoff of validation p-value of <0.005 . Although four of the 13 validated novel loci were identified in a previous GWAS: Rv2752c, whiB6, ubiA and intergenic region upstream of thyX, only the last two were consistently identified, i.e. in reported by more than one study. We believe GWAS results are strengthened by consistency and agreement with prior literature. Further neither of these loci have been accepted, or are currently used for resistance diagnosis or genome based prediction (see Mykrobe resistance predictor <https://github.com/iqbal-lab/Mykrobe-predictor> and <http://tbdr.lshtm.ac.uk/>). Further we identify 9 loci not previously described in the literature. We discuss available evidence on the association of the candidate loci with resistance in the discussion section by drug and also provide a meta-analysis table (supplementary Table 8).

R2.10 Discussion

The discussion needs to be revised according to the points made before. There actually not a lot that is discovered new. And the relevance for resistance development is not really clear. This

should be put in a sensitive discussion. The MIC data are not really relevant for the major analyses performed. So, I am not so sure that these data really show. This represent an interesting part of the work, but needs to be better uses.

A2.10 We respectfully disagree with the reviewer. We find and rigorously validate nine loci not previously described in the literature, this was in no doubt supported by the quantitative measurement of resistance that increased the power of the analysis. We are also the first to measure resistance phenotype heritability in TB, this would not have been possible without the measurement of MICs. We also demonstrate gaps between heritability attributable to known resistance loci and total heritability. We think this has strong implications for resistance prediction including likely the need to incorporate genome wide and lineage defining variants in resistance prediction. Further, our results also highlight the role of intergenic regions and demonstrates for the first time that promoter variants have consistently lower effects on MIC than gene body variants. This is also supported by impressions provided by the first and third reviewer. We have now altered the discussion throughout to highlight these points more clearly.

R2.11 The variants in the intergenic regions need to be more thoroughly investigate and, as they are only few, may be confirmed by a secondary method e.g. PCR and sequencing.

A2.11 The MTB genome is only 10% noncoding, and the noncoding portion of the genome is not more likely than the coding genome to harbor repetitive regions or regions that are difficult to sequence using next generation sequencing (NGS). Also two prior papers Beck TG et al Clin Chem 2016 (human DNA) and Feuerriegel et al JCM 2015 (MTB DNA) provide evidence against the need to validate Illumina based SNP calls using Sanger sequencing and even raise concerns about Sanger's false negative rate. In the Beck et al paper a mutation captured by NGS and not by Sanger was much more likely to be a false negative Sanger call than a false positive Illumina/NGS call. Further we found the associated non-coding variants in many isolates as detailed in Supplementary table 5, further reducing the likelihood that these are related to false positive NGS calls. We are thus confident in the accuracy of the variant calls in these regions, and unfortunately given the duration of time it takes to grow MTB in culture, re-extract DNA and re-sequence, attempting this additional step will incur considerable delays that will threaten the novelty of our work.

Reviewer #3 (Remarks to the Author):

This study performed a GWAS on 1,452 clinical MTB isolates to evaluate genome wide associations between mutations in MTB genomic loci and drug resistance. The major findings of this paper are the novel associations between 13 genomic loci and drug resistance, validated using an independent dataset, and estimates for the heritability of resistance phenotypes to 11 anti-TB drugs.

The initial GWAS identified 50 loci associated with increased resistance to one or more antibiotics. In lieu of functional validation of these variants, the authors used an independent dataset to attempt to validate the mutations identified in the initial GWAS and confirmed 13 of these located in genes along with to intergenic regions.

R3.1 A major difficulty with bacterial GWAS generally is the ability to accurately account for population structure and I am satisfied, based on the provided QQ plots, that the authors have done this. The details in the methods section, particularly those to do with GEMMA, should be expanded so as to make it possible to easily reproduce these results (reproducibility, to date, is another major problem in GWAS generally). It would be helpful to include a file with the commands used for all the analyses, not just the GWAS, in a supplementary file or else make the code available online through GitHub.

A3.1 We have now added an R and bash files containing all the commands that were executed on the input data. We also provide all input data necessary to execute the analysis and build the figures and tables.

Apart from this, the only other changes I'd suggest are:

General changes:

R3.2 Ensure consistency throughout for how numbers are displayed (1,526 versus 1526)

A3.2 We have now reviewed all numbers and convert all to a format that includes a comma (e.g. 1,526)

R3.3 Use 'and' instead of &

A3.3 We have made this edit throughout

R3.4 Check the use of 'where' and 'were'

A3.4 We have reviewed and made corrected any errors found

R3.5 Use words instead of numbers for numbers less than ten

A3.5 We have made this edit throughout

R3.6 The paper could do with a general proof read as there are a few minor typos and grammatical errors included.

A3.6 We have now proof read the paper and made adjustments.

Specific changes:

Abstract

R3.7 Add MIC and WGS in brackets after minimum inhibitory concentration and whole genome sequencing respectively

A3.7 We have now made both additions.

Introduction

R3.8 Remove ‘grim’ from the sentence “..resulting in the grim reality of...”. This seems too conversational to me

A3.8 We have now removed the words “the grim reality of”. The sentence now reads as follows: “The World Health Organization (WHO) estimates that only two of every three patients with multidrug resistant TB are diagnosed, three in every four of the diagnosed are treated, and only one of every two of the treated patients are cured, resulting in about 75% of the incident cases persisting in the community or succumbing to their illness.”

R3.9 Please provide reference for the statement ending “...MTB critical concentrations are largely based on consensus and lack solid scientific support.”

A3.9 We have now tempered this sentence and added references to Schon et al JAC 2009 and the recent WHO technical report on drug susceptibility testing of MDR TB (citations 14 and 15).

“Although such ‘binary’ DST is currently the standard to guide patient care, MTB critical concentrations lack consistent scientific support and several are based only on consensus^{3,4}.”

R3.10 Give examples of second line drugs in sentence ending “...ethambutol and second line drugs”

A3.10 We have now modified this sentence to read: “second line drugs including the injectable agents^{5,6}”

Results

R3.11 The use of the abbreviation SNV after single nucleotide substitutions; the ‘V’ stands for variant not substitution so change substitution to variant

A3.11 As we need to distinguish between substitutions and insertions/deletions (indels), and because technically the term variant encompasses both substitutions and indels, we have now replaced SNV with SNS: single nucleotide substitution throughout.

R3.12 Provide a meaning for AF (allele frequency?)

A3.12 We have now added a definition in the result section for AF in the following sentence: “The allele frequency *i.e.* the frequency of the minor variant within our sample, was <10% (Table 1, Table S2) in all but two validated loci.”

R3.13 I suggest providing a slightly more detailed explanation of what a ‘good’ QQ plot should look like for those readers unfamiliar with them (this author has seen much worse than yours!)

A3.13 We have now added a sentence to elaborate on the QQ plot assessment in the result section as follows “QQ plots of the resultant p-value distribution suggested that the correction for population structure was adequate. This is demonstrated by the adherence of the observed p-value distribution to the expected line with the exception of the short tail indicating the significant loci in Supplementary Figures 2 and 3.”

Discussion

R3.14 Provide a reference for the statement ending “..were previously associated with resistance in a prior GWAS albeit to non-AG agents”

A3.14 We apologize about this oversight. The reference to Zhang et al. Nature Genetics 2013 has now been added (citation 12).

I believe that the results are novel and of potential interest to the wider field and, as far as I’m aware, is the first study to use GWAS to identify genes and regulatory regions associated with particular MIC values in MTB.

Reviewer #1 (Remarks to the Author):

The authors have addressed all my comments. I congratulate them for the effort done

Reviewer #2 (Remarks to the Author):

The authors have addressed the majority of comments raised. I have no further suggestions

Reviewer #3 (Remarks to the Author):

Thank you for addressing my concerns and making changes where required. I am also satisfied that the authors have made an attempt to adequately deal with the requested changes from the other reviewers.

On this basis, I believe the paper merits publication in Nature Communications.

REVIEWERS' COMMENTS:

Reviewer #1 (Remarks to the Author):

The authors have addressed all my comments. I congratulate them for the effort done

Reviewer #2 (Remarks to the Author):

The authors have addressed the majority of comments raised. I have no further suggestions

Reviewer #3 (Remarks to the Author):

Thank you for addressing my concerns and making changes where required. I am also satisfied that the authors have made an attempt to adequately deal with the requested changes from the other reviewers.

On this basis, I believe the paper merits publication in Nature Communications.